# Impactor material records the ancient lunar magnetic field in antipodal anomalies

S. Wakita [1,2✉], B. C. Johnson [1,3], I. Garrick-Bethell [4,5], M. R. Kelley[4], R. E. Maxwell[4] & T. M. Davison [6]

The Moon presently has no dynamo, but magnetic fields have been detected over numerous portions of its crust. Most of these regions are located antipodal to large basins, leading to the hypothesis that lunar rock ejected during basin-forming impacts accumulated at the basin antipode and recorded the ambient magnetic field. However, a major problem with this hypothesis is that lunar materials have low iron content and cannot become strongly magnetized. Here we simulate oblique impacts of 100-km-diameter impactors at high resolution and show that an ~700 m thick deposit of potentially iron-rich impactor material accumulates at the basin antipode. The material is shock-heated above the Curie temperature and therefore may efficiently record the ambient magnetic field after deposition. These results explain a substantial fraction of the Moon's crustal magnetism, and are consistent with a dynamo field strength of at least several tens of microtesla during the basin-forming epoch.

[1] Department of Earth, Atmospheric, and Planetary Sciences, Purdue University, West Lafayette, IN 47907, USA. [2] Department of Earth, Atmospheric and Planetary Sciences, Massachusetts Institute of Technology, Cambridge, MA 02139, USA. [3] Department of Physics and Astronomy, Purdue University, West Lafayette, IN 47907, USA. [4] Department of Earth and Planetary Sciences, University of California, Santa Cruz, CA 05064, USA. [5] School of Space Research, Kyung Hee University, Yongin, Gyeonggi 446-701, Korea. [6] Department of Earth Science and Engineering, Imperial College London, London SW7 2AZ, UK. ✉email: swakita@purdue.edu

Since the Apollo era, spacecraft have detected strong localized magnetic fields coming from the lunar crust, known as magnetic anomalies[1,2]. These anomalies record ancient magnetic fields that may have arisen from a dynamo[3,4] or possibly from dynamo fields amplified by large impacts[5], but not likely from fields produced by impact amplification of the interplanetary magnetic field[6]. However, an impediment in using magnetic anomalies to study the ancient lunar field is that their geologic formation mechanism is not known. Hypotheses for their formation include intrusive volcanism[7], crater melt sheets[8], ejecta deposits[1,9–11], and cometary interactions with the surface[12]. The antipodes of the Imbrium, Serenitatis, Orientale, and Crisium basins all exhibit large provinces of crustal magnetism[5,13,14] and in some cases modified terrain[5,15,16]. A hypothesis for the formation of these provinces is that basin ejecta may have accumulated at the basin antipode, which then became magnetized by cooling in an ambient field. Pressure remanent magnetization of the rock upon landing is unlikely due to the inefficiencies of this process[17], the low-peak shock pressures (<12 GPa) from ejecta landing at ~2 km/s[18], and the randomization of rock-scale remanence directions after initial surface contact.

Here, we explore the hypothesis that antipodal ejecta contains sufficient impactor material to explain the observed magnetization of anomalies antipodal to large basins, using high-resolution impact simulations. Previous exploration of antipodal ejecta deposits did not explore the fate of impactor materials[5]. For moderately oblique impacts we find that antipodal ejecta is dominated by the impact materials, which can have high thermoremanent magnetization (TRM) susceptibility ($X_{TRM}$) like the chondritic meteorite. Moreover, this ejecta is above the Curie temperature at the time of emplacement and can thus record the magnetic field of the Moon as it cools.

## Results and discussion

**The lunar magnetic field for the antipodal magnetic anomalies.** To date, the major difficulty with the antipodal ejecta hypothesis has been that all lunar materials so far discovered are not sufficiently magnetic to produce the fields observed at spacecraft altitude, for realistic values of the ancient magnetizing field. For example, let us assume Crisium-antipode anomalies were formed from lunar materials with the highest $X_{TRM}$ found in the Apollo sample collection, $X_{TRM} = 0.01$ (i.e., lunar impact melt breccias[11]). Focusing on the strongest anomalies within the antipode region, which are at the lunar swirls (Fig. 1a), we assume a magnetized layer thickness comparable to the horizontal length scale of the swirl pattern[19]. A forward model using these assumptions produces the observed 50 nT magnetic field anomaly at 20 km altitude with an ancient magnetizing field of ~500 μT (Methods subsection The strongest magnetic anomalies at the Crisium antipode, Supplementary Fig. 1), which is ~10 times the Earth's present field. Such high inferred dynamo field strengths of 500 μT are likely untenable, since current theory suggests that lunar paleofields up to only ~3–15 μT may be achievable[4,20]. Even the paleofield values inferred from most strongly magnetized Apollo samples are ~5–10 times weaker than 500 μT[4]. As the inferred paleofield strength is inversely proportional to assumed $X_{TRM}$, our forward model implies material at the Crisium antipode with a $X_{TRM}$ ~30–150 times higher than lunar materials would be consistent with the ~3–15 μT dynamo field limits predicted by theory. Wieczorek et al.[11] use similar reasoning to argue that the magnetized materials peripheral to the South Pole-Aitken basin are not indigenous to the Moon. Rather, this material may have been provided by meteoritic impactor material, where $X_{TRM}$ may in fact be up to 100 times higher than lunar materials[11]. If

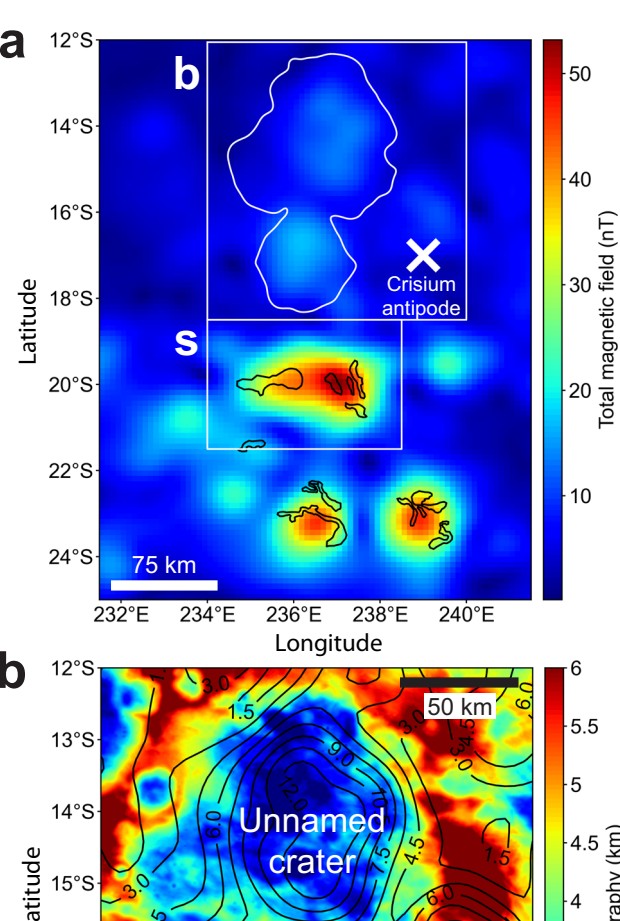

**Fig. 1 Family of magnetic anomalies antipodal to the Crisium basin.**
**a** Total magnetic field map at 20 km altitude (magnetic data are from the model of Tsunakawa et al. 2015[2]). Box s shows the region with the strongest observed crustal field on the Moon. Black outlines show locations of bright albedo anomalies known as lunar swirls; white outlines show topography contours of craters in panel b. **b** Zoom of box b in panel **a**. Total magnetic field contours (black lines) over topography (topography data are from Smith et al. 2010[65]). Field contours follow the crater topography in Houzeau crater, and in the eastern half of the unnamed crater, suggesting crater fill deposits are the source of the magnetized material. The contours are from 1.5 nT to 18 nT, in 1.5 nT intervals. Topography is saturated at 2 and 6 km to highlight crater rim topography.

the Moon's antipodal ejecta deposits formed from such extralunar meteoritic material, it would explain a large fraction of lunar crustal magnetism.

**Observational evidence for ejecta magnetization hypothesis.** To illustrate the viability of the antipodal ejecta magnetization hypothesis, we examine a family of magnetic anomalies at the Crisium basin antipode, which exhibit the strongest magnetic

fields on the Moon (Fig. 1 and Supplementary Fig. 2a). We find that the two northernmost anomalies have magnetic fields that are correlated with crater interior topography (Fig. 1a, b). One of these craters, Houzeau, is substantially shallower than similarly sized craters elsewhere on the Moon, suggesting it filled with material after its formation (see also Methods subsection Crater fill deposits at Houzeau crater near the Crisium antipode and Supplementary Fig. 3). For example, we compared Houzeau's azimuthally averaged topography to the craters Jackson (22.1° N, −163.3° E) and Einthoven (−5.1° N, 110.0° E) (Supplementary Fig. 4). Houzeau's stratigraphic age is undefined, while the stratigraphic ages of Jackson and Einthoven are Copernican and Nectarian, respectively[21]. Despite the differences in their ages, the floor elevations of Jackson and Einthoven are similarly lower than Houzeau by 1.3–1.5 km (note that Jackson has a central peak while Einthoven does not; our elevation difference estimate allows for the central peak topography). This suggests there is 1.3–1.5 km of post-formation fill in Houzeau, which may be ejecta from the Crisium basin, and which may be responsible for its topographically-correlated magnetic anomaly. Finally, the Crisium basin itself exhibits magnetic anomalies in its interior, consistent with a strong, long-lived magnetic field shortly after its formation[22].

**Antipodal ejecta from impactor.** We find the occurrence and distribution of antipodal ejecta strongly depends on the impact angle. For a 100-km-diameter impactor striking at 12 km/s at 20°, 30°, and 45° (Fig. 2a, b) ejecta is preferentially distributed downrange and some ejecta reach the impact antipode. However, more vertical impacts produce much smaller antipodal ejecta deposits: only two tracer particles within 3° of the antipode for 60° impacts and none within 40° of the antipode for 90° impacts (Fig. 2c, d). A similar trend occurs for impacts at 17.4 km/s (Supplementary Fig. 5). This finding is consistent with previous work[5]. Note that a 100-km-diameter impact forms a ~1000-km-diameter basin, similar to the diameter of the Crisium basin (Methods subsection Numerical Methods, Supplementary Note 2). As the impact angle becomes more oblique, the impact ejects material at a lower ejection angle. Thus, oblique impacts are more suitable for producing antipodal ejecta than impacts closer to vertical incidence. As expected, antipodal ejecta arrives later than more proximal ejecta. The earliest ejecta from a 45° impact landing within 3° of the antipode arrives in ~10 h (Fig. 3 and Supplementary Fig. 6) and the flight time of 77% of the ejecta is within 30 h. Trajectories of antipodal ejecta from lower impact angles are closer to the Moon's surface. These shorter flight paths result in smaller flight times (Fig. 3 and Supplementary Fig. 6); 93% ejecta produced by the 30° impact land within 3° of the antipode within 10 h.

Most importantly for this study, we find that antipodal ejecta is composed of both impactor and target material (Fig. 2a, b). The ejecta sourced from the Moon originates from relatively shallow depths (up to 20 km). Antipodal ejecta from the impactor are sourced from the impactor's outermost 35 km (Supplementary Fig. 7). If lunar magnetic anomalies are produced by magnetized impactor material we might expect them to have a distinct composition. Remote sensing studies, however, have never detected a difference in soil composition at magnetic anomalies compared to surrounding rock. For example, at the Crisium antipode, we find no correlation between magnetic field and soil iron content (Supplementary Fig. 2). This can be explained by the fact that ejecta sourced from the Moon tend to have longer flight times and will bury earlier arriving impactor material (Fig. 3 and Supplementary Fig. 6).

**Capability of antipodal ejecta to record ancient magnetic fields.** During its flight, antipodal ejecta must stay near the Curie temperature until emplacement on the surface to record a strong TRM in the ancient lunar magnetic field. The cooling of fragments is limited by the rate of conduction and the cooling timescale of ejecta is $\tau_{cool} \approx d^2/\kappa$, where $d$ is their size and $\kappa = 10^{-6}$ m$^2$/s is the thermal diffusivity[23]. Recent simulations of impact fragmentation suggest that ejecta capable of reaching the antipode produced by a 100-km-diameter impactor will consist of 50–1000 m scale fragments[24]. These estimates are consistent with extrapolation from observations of secondary craters of Orientale[25]. Even ejecta fragments as small as 0.5 m have a cooling timescale, $\tau_{cool} = 70$ h, which is longer than the flight time of most antipodal ejecta (>93%). Thus, for an impact of this scale, we expect most solid ejecta heated to near the Curie temperature would be able to record TRM.

When impacts eject highly shocked materials, they may melt. Most melt droplets produced by an impact are likely mm in scale with the largest melt particles several cm in scale[26,27]. The cooling timescale of $10^{-3}$–$10^{-1}$ m melt particles is $\tau_{cool} = 1$ s to 2.7 h, which is shorter than the flight time of antipodal ejecta. Thus, any ejecta that was once molten would likely be too cool to record the lunar magnetic field after emplacement on the surface. Our simple calculation, however, does not account for radiation from nearby melt particle, which could extend this time[28]. The antipodal ejecta produced during impacts at 17.4 km/s experience higher peak shock pressures and more melting than those produced by impacts at 12 km/s (Fig. 3 and Supplementary Fig. 8); 89% and 15% of antipodal ejecta from the 45° impacts is unmelted for impact velocities of 12 and 17.4 km/s, respectively. The 12 km/s impact simulations produce more unmelted antipodal ejecta exceeding the Curie temperature, indicating that lower impact velocities are more conducive to producing magnetic anomalies.

**Implications for the lunar magnetic field.** It is important to determine that enough impactor material arrives at the antipode to produce a magnetic anomaly observable from lunar orbit, for realistic values of the ancient magnetizing field. The total magnetic moment required to produce the field at 20 km altitude at the strongest anomalies near Crisium's antipode (the swirls in Fig. 1a, box s) is ~$1.3 \times 10^{13}$ Am$^2$ (Methods subsection The strongest magnetic anomalies at the Crisium antipode, Supplementary Fig. 1). The 12 km/s and 17.4 km/s impacts at 45° produce 695 m and 809 m of impactor material at the antipode, respectively (Fig. 4, Supplementary Fig. 9, Supplementary Note 3, Supplementary Table 1). Using 700 m as a representative thickness, assuming the rock magnetization $M$ is concentrated under the high-albedo swirls in this region, and taking their surface area to be $6.4 \times 10^8$ m$^2$, we find $M = 29$ A/m. This value is consistent with values estimated for the Moon's Reiner Gamma magnetic anomaly (70 A/m), and is similar to the strongest anomalies on Earth (>100 A/m [29]). Reiner Gamma's peak field strength at 20 km altitude is also similar to that at the Crisium antipode (Fig. 1a)[8]. If the impactor was similar to a parent body of chondritic meteorites, $X_{TRM}$ may range from ~0.1–1.5, with a mean of $0.5 \pm 0.4$ (1σ), based on 15 meteorite classes in Table S3 of Wieczorek et al.[11]. Since impactor ejecta is sourced from multiple depths within the impactor (Supplementary Fig. 7), the $X_{TRM}$ values of the antipodal ejecta may be diverse. This diversity may explain the variable distribution of magnetic field strength in the antipodal magnetized province; e.g., the highest magnetic field values are found at swirls (Fig. 1a). The diversity in $X_{TRM}$ in meteoritic material, and the diverse differentiation states of impactor bodies, may also explain why some basins form

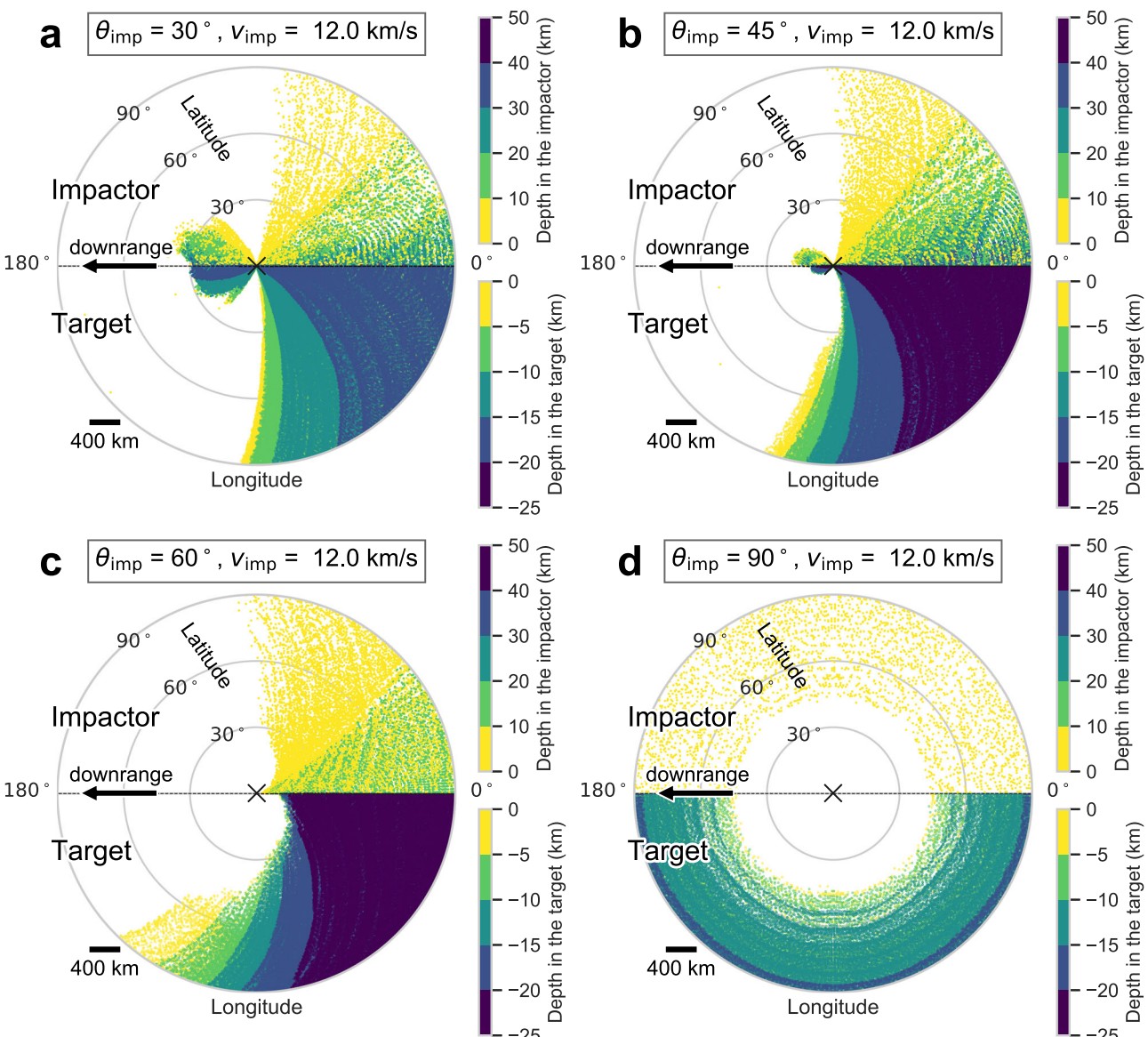

**Fig. 2 Spatial distribution of ejecta landing on the antipodal hemisphere.** Each panel represents ejecta from (**a**) 30°, (**b**) 45°, (**c**) 60°, and (**d**) 90° impacts with 12 km/s, respectively. Top halves of each panel show ejecta sourced from the impactor and colored according to the depth in the impactor. Bottom halves depict ejecta sourced from the target and colored according to the depth in the target. The black cross symbols at the center represent the antipode of the impact site. Ejecta from oblique impacts distribute in a downrange direction (as indicated by an arrow); ejecta which pass the antipode are shown in left side in **a** and **b**. Note that some ejecta overlap; ejecta from 5 km depth are plotted underneath ejecta from much deeper than 5 km.

antipodal magnetic anomalies, while others do not. Large impactors like those considered here may be intact parent bodies or represent a piece of an even larger parent body. The formation time of the parent body helps determine whether it is differentiated or not[30,31], which may result in a diversity of $X_{TRM}$ values. When a parent body forms within 1–2 Myr after the birth of the Solar System, it may fully differentiate, like the parent bodies of iron meteorites[32]. Later formed bodies could remain undifferentiated like a parent body of chondrites[30,33]. The parent body could also be partially differentiated, possibly producing both iron meteorites[34] and chondrites[35–37]. If we take the $+1\sigma$ -$X_{TRM}$ value of 0.9 to represent the more strongly magnetized materials, and assuming $M = 29$ A/m, we infer the estimated ancient lunar magnetic field $B$ to be 40 µT. If we take the mean, $X_{TRM} = 0.5$, we obtain 73 µT. Either of these values are consistent with fields estimated to have existed during the high-field epoch from ~3.5–3.9 Ga, based on studies of lunar samples[4].

The impactor material found at basin antipodes can explain some of the Moon's enigmatic magnetic anomalies. Similar analyses as above may obtain the minimum paleofield strengths that magnetized these regions, depending on the thicknesses of materials deposited there. Since the ages of some basins can be estimated from their crater density statistics, the ages of the magnetizing fields can be inferred, a capability that so far has been limited to studies of Apollo samples. We ran our simulation of a 100-km-diameter impactor with 45° at 12 km/s at lower resolution until the final crater has formed. This simulation indicates that, in addition to producing antipodal anomalies, shock-heated impactor materials will also be distributed in and around large basins (Supplementary Note 4 and Supplementary Figs. 10 and 11). These results indicate that an impactor material from a basin-forming impact can produce antipodal anomalies as well as the magnetic anomalies found in and around large basins (e.g., Imbrium[38], South Pole-Aitken basin[11], and Crisium[22]). If

the composition of impactor material at magnetic anomalies can be determined, this would provide information about the impactors that bombarded the inner solar system. Moreover, $X_{TRM}$ measurements of such material would ultimately yield more robust estimates of the strength of the ancient lunar magnetic field and vigor of the Moon's dynamo.

## Methods

**The strongest magnetic anomalies at the Crisium antipode.** In the main text we argue that unrealistically strong fields would be required to produce the magnetic fields arising from magnetized rocks at antipodal regions at spacecraft altitudes, assuming known lunar materials. To demonstrate this, we start by modeling the strongest anomalies at the Crisium antipode, i.e., the swirls in Fig. 1a box s, as two-

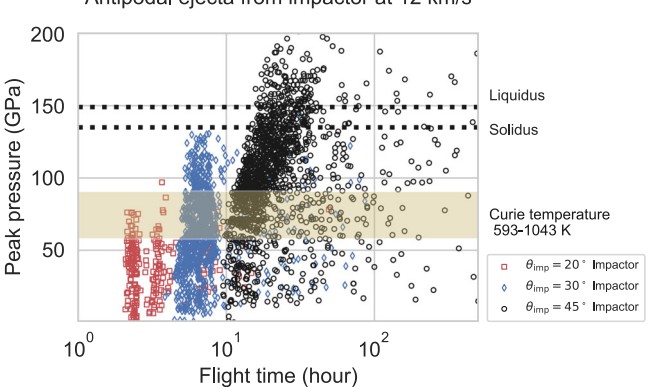

Antipodal ejecta from impactor at 12 km/s

**Fig. 3 Peak pressure of antipodal ejecta sourced from the impactor as a function of their flight time.** Each symbol represents ejecta from 12 km/s impacts with different impact angles (see legends). Note that we only plot the antipodal ejecta located within 3° of the antipode. Horizontal beige bar (57–90 GPa) shows the corresponding pressure of the Curie temperature of pyrrhotite (593 K) and iron (1043 K) based on the equation of state of dunite[57]. We take pyrrhotite as a lower limit of Curie Temperature. Pyrrhotite is found in carbonaceous chondrites, possible impactors, as a remanence carrier[66]. Horizontal black dashed lines denote solidus (incipient melting) and liquidus (complete melting) of dunite[67].

dimensional sheets of dipoles (Supplementary Fig. 1a, b). The sheets were generated to take on the same surface area that comprises the swirls, which assumes that the swirls contain the majority of the magnetized material. This assumes the bright soil at swirls is at least partially the result of the magnetic field blocking full solar wind access to the surface[39–43]. Hence, swirls will have stronger magnetization than adjacent terrain without swirls. We use the mapping of Denevi et al.[44] to identify the boundaries of the swirls throughout this paper.

In our modeling, we use a magnetization direction for the anomalies in Supplementary Fig. 1a using a technique proposed by Parker et al.[45,46]. Briefly, the method assumes that the magnetization is unidirectional and distributed throughout the subsurface. In this case the problem can be reduced to modeling a sheet of dipoles at the surface of variable strength and equal direction. This inversion applied to the entire region in the white box s in Fig. 1a yields a minimum error solution of declination −10° and inclination 51° (using a source-dipole grid-spacing of 0.1 degrees; initial results reported by Maxwell and Garrick-Bethell[47]. As the swirls are shorter wavelength features than the spatial sampling resolution of the magnetometers, and are also smaller than the sampling altitude, we do not use the results from Parker's method to map a best-fit two-dimensional distribution of magnetized material. The results from such a method are too coarse. Instead we use the sheet distribution correlated with the swirl locations, discussed above, with the best-fit direction from Parker's method. Importantly, even extreme uncertainties in the magnetization direction would not affect our estimate of peak magnetizing field strength by a factor of more than two.

We also note that a preliminary analysis of the magnetization directions of several of the strongest anomalies at Crisium's antipode have similar directions[47], which supports the idea they were magnetized in a unidirectional dynamo magnetic field.

In the next step in our modeling we increased the total magnetic moment of the sheets of dipoles until the peak total field matched the observed peak total field at 20 km altitude (Supplementary Fig. 1b). This produced a total moment of $1.30 \times 10^{13}$ Am². We then assumed the magnetized layer thickness is equal to 5000 m, which is similar to the horizontal length scale of the swirls. This thickness approximation is motivated by previous work that showed the depth to the top of the source must be smaller than the horizontal length scale of the swirl, if swirls are indeed formed by standoff of the solar wind[19]. We note that smaller assumed layer thicknesses would require higher $M$, and therefore higher even less realistic magnetizing fields for indigenous lunar materials (see end of paragraph). Using the observed surface area of the swirls ($6.4 \times 10^8$ m²), we then calculated the volume of the magnetized material to obtain a bulk magnetization $M = 4$ A/m. We then related $M$ to the ambient field via (see discussion in Wieczorek et al.[11]):

$$M = X_{TRM} B/\mu_o \quad (1)$$

where $B$ is the ancient magnetizing field and $\mu_o$ is the magnetic constant ($4\pi \times 10^{-7}$ N/A²). Assuming $X_{TRM} = 0.01$, which is the highest value found in lunar samples (see main text; Wieczorek et al.[11]), we obtain $B = 500$ μT, which is much higher than any known theory of dynamo generation could produce on the Moon. The field would still be implausibly high (250 μT) even if the magnetization were spread over an area two times larger than the swirl borders. Hence, we conclude

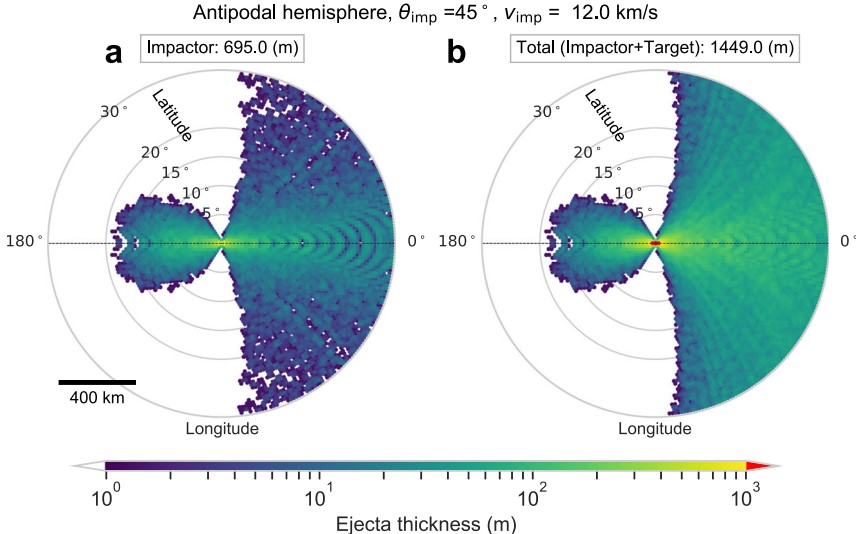

Antipodal hemisphere, $\theta_{imp}$ =45 °, $v_{imp}$ = 12.0 km/s

**Fig. 4 Thickness of ejecta from a 45° impact with 12 km/s within 30° in latitude from the antipode.** Panel **a** represents the ejecta sourced from the impactor and **b** is the total ejecta (both from the impactor and the target). The horizontal dotted line represents the direction of the impactor (see arrow and caption in Fig. 2). Note that each circled point represents the surface area of a spherical cap with a diameter of 1°, in which we sum the volume of ejecta and derive its thickness. The arc-shaped patterns on the right side of the antipode appear because of the numerical resolution and the choice of surface area size (Supplementary Note 3 and Fig. 9).

that exogenous (non-lunar origin) materials with higher $X_{TRM}$ values are likely to have acquired the magnetization at the Crisium-antipode magnetic anomalies.

**Crater fill deposits at Houzeau crater near the Crisium antipode**. In the main text we showed that the magnetic field morphology at Houzeau crater and the unnamed crater to its north are correlated with the crater interior topography (Fig. 1a, b). Here, we consider the possibility that some fraction of Houzeau's fill may be Orientale ejecta. The most recent geologic map of the Moon[21] shows that Houzeau and nearby swirls (strongly magnetic anomalies) are mostly within Orientale's outermost ejecta deposit (mapped as unit Ioho, Supplementary Fig. 3), with some minimal overlap of the inner ejecta deposit in the east (mapped as unit Iohi). Both of these units comprise parts of the Hevelius Formation. The interpretation of the Ioho unit is that it is the "Thinning distal margins of Orientale basin ejecta"[21]. We can estimate the thickness of the Orientale ejecta at Houzeau crater, which is located 900 km from the center of the Orientale basin. Hood et al.[5] estimated that for a 100-km-diameter impactor with impact angle of 45°, the ejecta thickness would be ~300–800 m at this distance (their Fig. 7b). However, Orientale is best modeled by a smaller impactor of diameter 64 km[48], suggesting that 800 m is a conservative upper limit on the contribution of Orientale material to Houzeau's fill. Ultimately, high-resolution ejecta modeling may be able to resolve the exact contribution, but for now, we can conclude that Orientale is unlikely to produce all of Houzeau's fill. Although the ejecta in our simulations cover a large area and the variation of ejecta are relatively smooth (Fig. 4), distal ejecta deposition is an inherently stochastic process, often complicated by heterogeneous contributions from rays[49], and we acknowledge that it cannot be modeled completely accurately with current simulations.

In the main text we also concluded that late arriving lunar ejecta at the antipode of a basin may bury the early-arriving impactor-derived ejecta, and thereby preserve the magnetism in the impactor material against micrometeoroid erosion over ~4 billion years. Such burial would also be consistent with the lack any reported compositional anomalies associated with magnetic anomalies. A veneer of Orientale ejecta may also help bury the impactor ejecta deposit.

**Numerical methods**. To determine the provenance and spatial distribution of antipodal ejecta we simulate impacts using the iSALE-3D shock physics code[50,51]. This code uses a solver as described in Hirt et al.[52] and includes a strength model[53–55]. As we are focused on ejecta antipodal to 1000-km-diameter scale basins, we simulate a 100-km-diameter impactor striking a flat target at a resolution of 1 km. Our resolution of 1 km or 50 cells per projectile radius is high enough to track the ejecta. Following Melosh et al.[56], we assume the spherical impactor and the flat target have homogeneous compositions of dunite[57]. Ordinary chondrites and S-type asteroids, which contain 20–30 wt% iron[58], are well represented by dunite[59]. Owing to the limitations of the current numerical code, we cannot consider differentiated impactors. Since our provenance plot indicates that the outer portion of the impactor dominates the antipodal ejecta (Supplementary Fig. 7), the antipodal ejecta can be considered to be mantle or undifferentiated material. We also consider the material strength and summarize input material parameters in Supplementary Table 2. We simulate impacts at 12 km/s and 17.4 km/s, 25% of the impacts on the Moon have impact velocities below 12 km/s and the latter represents the mean impact velocity[60]. To examine the dependence of the impact angles on the distribution of antipodal ejecta, we vary impact angles as 20°, 30°, 45°, 60°, and 90° (vertical), and trace their ballistic flight on a non-rotating Moon (see Supplementary Note 5 for discussion of the effect of rotation).

Our 100-km-diameter impactor simulations are valid for roughly 1000-km-diameter scale basins. Using the scaling law[61], a 45° impactor with 100 km in diameter at 12 and 17.4 km/s produces basins 930 and 1120 km in diameter, respectively. For comparison Crisium is 1076 km in diameter[62]. Assuming hydrodynamic similarity, the ejecta thicknesses reported here can also be scaled linearly by impactor size[63] (see also Supplementary Note 2).

To track ejecta properties, such as position, velocity, and pressure, we place a Lagrangian tracer particle in each computational cell at the beginning of impact simulations. Once tracer particles as ejecta reach a height of 50 km, we use their position, speed, and ejection angles to solve the analytical equation for the trajectory of ejecta[64]. To reach the antipode of the lunar basin, the ejecta velocity should range between the escape velocity of the Moon (2.34 km/s) and a circular orbit (1.68 km/s). Since we assume the ejecta travel over a non-rotating target, the ejection angle must be less than 45° to land at the antipode.

## Data availability
The datasets generated during and/or analyzed during the current study are available from the corresponding author on reasonable request. Lunar magnetic field data are from the model of Tsunakawa et al. (2015)[2] and can be found at http://www.geo.titech.ac.jp/lab/tsunakawa/Kaguya_LMAG. Lunar topography data can be found at https://pds-geosciences.wustl.edu/lro/lro-l-lola-3-rdr-v1/lrolol_1xxx/DATA/LOLA_GDR/. Lunar iron abundance data can be found at: https://pds-geosciences.wustl.edu/missions/lunarp/reduced_special.html. Our simulations data are given by using iSALE-3D, and our input files are available on the website https://doi.org/10.5281/zenodo.5548201.

## Code availability
The code to determine the magnetization direction is available from the authors upon reasonable request. Usage of the iSALE-3D code is restricted to those who have contributed to the development of iSALE-2D, and iSALE-2D is distributed on a case-by-case basis to academic users in the impact community. It requires a registration from the iSALE webpage http://www.isale-code.de, and usage of iSALE-2D and computational requirements are also shown there. To analyze the binary data from iSALE-3D and plot figures, we use pySALEPlot, which is included in iSALE-3D. Please note that pySALEPlot in iSALE-2D would not work for the data from iSALE-3D.

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

## Acknowledgements

We gratefully acknowledge the developers of iSALE-3D, including Dirk Elbeshausen, Kai Wünnemann, and Gareth Collins. This research was supported in part through computational resources provided by Information Technology at Purdue, West Lafayette, Indiana. I.G.B. acknowledges support from the NASA Lunar Data Analysis program, and the BK21 plus program funded through the National Research Foundation (NRF), funded by the Ministry of Education of Korea. T.M.D. was funded by STFC grant ST/S000615/1.

## Author contributions

S.W. conducted the formal analysis of the simulations and wrote the original draft with contributions from coauthors. B.C.J. and I.G. gave conceptualization and methodology. I.G., M.R.K. and R.E.M. analyzed the magnetic properties of the antipodal material. T.M.D. developed the software, iSALE-3D and pySALEPlot. All authors contributed to manuscript preparation and the conclusions presented here.

## Competing interests

The authors declare no competing interests.
