## [Peer Review File · Nature Communications]

Impactor material records the ancient lunar magnetic field in antipodal anomaliesEditorial Note: Parts of this peer review file have been redacted as indicated to avoid any copyright infringement.

Reviewer Comments:

Reviewer #1 (Remarks to the Author):

This manuscript presents a combination of numerical impact simulations and modeling of magnetic anomalies antipodal to the lunar Crisium basin to argue that the observed concentrations of magnetic anomalies antipodal to young large lunar basins can be explained as due to unusually thick concentrations of ejecta containing ferromagnetic material (i.e., iron) from the impactors that created the basins. The numerical simulation results are similar in many ways to those presented previously by Hood and Artemieva (2008), e.g., showing that oblique (less than 45 degrees from horizontal) impacts are needed to produce antipodal ejecta deposition and that the thickness of the deposits is largest at the antipode. The expectation of antipodal concentrations of ejecta goes back at least to ballistic calculations presented by H. J. Moore et al. (Proc. Lunar Sci. Conf. 5th, p. 71, 1974). However, these numerical simulations are more detailed than those reported previously and add new information. For example, the impactor or target origination of the ejecta is shown (Figure 5), allowing an explanation for why the surface composition in basin antipode zones is not more iron-rich than surrounding regions (i.e., ejecta originating in the target arrives later and buries ejecta originating in the impactor). Also, the modeling of anomaly sources antipodal to Crisium shows more clearly that the amplitudes of observed anomalies probably requires stronger magnetization intensities than can be expected for the most ferromagnetic returned lunar samples (i.e., impact melt breccias). This implies that additional ferromagnetic material (probably iron) from the impactor is needed to explain the anomalies.

The issue of whether the observed concentrations of magnetic anomalies antipodal to young large lunar basins implies a physical connection to the basins or is merely coincidental is a long-standing issue of lunar science. Lin et al. (1988; referenced here), using an early large-scale map of lunar crustal fields, presented some simple statistical arguments that the observed strong anomaly concentrations antipodal to four of the youngest large lunar basins (Imbrium, Orientale, Serenitatis, Crisium) are unlikely to have occurred by chance. Mitchell et al. (Icarus, v. 194, p. 401, 2008) presented more detailed Monte Carlo simulations using a better large-scale map to draw a similar conclusion. However, the anomalies antipodal to Imbrium and Serenitatis are also concentrated along the northwestern periphery of the relatively ancient South Pole-Aitken (SPA) basin. This led to the proposal by Wieczorek et al. (2012) that iron from the impactor that created this basin was responsible for the observed anomalies. In fact, the latter reference presented numerical impact simulations and magnetic anomaly modeling peripheral to an impact basin that were similar in many ways to those presented in this manuscript for basin antipode zones.

But evidence for associations of anomalies with young large basins persisted. A later study found a concentration of anomalies near the antipode of the polar Schrödingier basin, which is one of the three youngest lunar basins (Hood et al., JGR Planets, v. 118, doi:10.1002/jgre.20078, 2013). The same study found further evidence for landform modification antipodal to young large basins, which may be consistent with antipodal ejecta deposition. The original proposal that ejecta from young basin-forming impacts are the main sources of orbital magnetic anomalies dates back to the work of

D. W. Strangway et al. (Nature Physical Science, v. 246, p. 112, 1973). Most recently, continued mapping of lunar fields has found evidence for elongated anomalies aligned radial to the Imbrium basin, (Hood, Fall AGU Meeting, 2019). Great circle paths determined by the orientations of the elongated anomalies pass through strong isolated anomalies and intersect in the Imbrium antipode zone where the largest group of strong anomalies is located (Hood et al., under review at JGR Planets at the time of peer-review of the present manuscript – now available at <https://doi.org/10.1029/2020JE006667>).

[redacted]

The current manuscript is therefore a welcome addition to the literature. It builds on the work of Wieczorek et al. (2012) who first showed quantitatively that impactor material (iron) in ejecta from a large basin-forming impact could potentially explain many lunar magnetic anomalies if this material cooled in the presence of an internal magnetizing field. It extends their impact simulations and modeling to the antipodes of the youngest large basins, supporting previous results (e.g., Hood and Artemieva, 2008; Mitchell et al., 2008). However, unlike these latter works, it explains most of the largest concentrations of anomalies as being due to antipodal deposition of ejecta enriched in ferromagnetic material (iron) from the impactors that created the basins. It remains possible that some of the anomalies north of SPA originated at the time of that ancient impact. But the present work shows that a majority of anomalies are more likely to date from the times of the youngest large basin-forming impacts.

I have relatively minor criticisms and questions for the authors to consider in their revision.

(1) The numerical simulations described here assume that both the Moon and the impactor have homogeneous compositions of dunite (mostly olivine). Since the main ferromagnetic carrier in lunar materials is Fe or Fe-Ni, a more realistic approach would be to consider an iron-rich impactor (either with distributed iron or iron in a differentiated core). This was the approach taken by Wieczorek et al. (2012). Can the authors explain briefly in the manuscript why a pure silicate composition is used? The Wieczorek et al. (2012) simulations, for example, tracked the mixing of iron into ejecta and impact melt. The current simulations show that impactor material from different depths in the 100 km diameter impactor ends up as ejecta in the antipodal zone (e.g., Figure 2). This is valuable but there is no explicit determination of how much impactor

iron is deposited in the antipodal zone.

(2) The presented impact simulations show that, for impact angles less than about 45 degrees from the horizontal, concentrated antipodal ejecta deposition occurs (Figures 2 and 4). This agrees with earlier simulations by Natalia Artemieva in Hood and Artemieva (2008). However, the simulations of Wieczorek et al. (2012) found that this range of impact angles also caused all ejecta containing iron from the impactor to be deposited either at the downrange edge of the basin inner rim or outside of the inner rim (see, e.g., their Figures S3A, S3B, and S3C). Crisium has magnetic anomalies well within its inner rim but also concentrated at the antipode. The anomalies within its inner rim suggest a near-vertical impact while the anomalies at the antipode suggest an oblique impact. Please add a few sentences to the text noting this inconsistency and that future simulations tracking the fate of impactor iron are needed to address it. Is it possible that an impact angle close to 45 degrees could produce anomalies both within the inner rim and at the antipode?

(3) Unusual modified landforms (referred to by geologists as ``material of grooves and mounds' and ``hilly and furrowed material') is found antipodal to young lunar basins, especially Imbrium and Orientale (see lunar geologic maps of these regions dating from the 1970s; for a review, see Hood et al., JGR, 2013). Similar unusual terrain is found antipodal to the Caloris basin on Mercury. Shock effects of converging ejecta impacts are one possible origin. No such terrain is found antipodal to Crisium, possibly because it is overlain by ejecta from the nearby Orientale basin. The existence of such terrain should at least be mentioned because it provides further evidence that antipodal effects of lunar basin-forming impacts are real and are most easily identified for the youngest basins. Is it possible that shock effects of converging ejecta impacts could have contributed to magnetization acquisition in these zones (in addition to TRM)?

(4) Previous work by one of the authors (Nayak, Hemingway, and Garrick-Bethell, Icarus, v. 286, p. 153, 2017) found a diverse set of magnetization directions for anomaly sources along the northern edge of SPA (Imbrium antipode zone). The most probable sources in that paper were considered to be igneous intrusions, which could have formed in different magnetic epochs. The directional incoherence was therefore attributed to ``surprisingly large amounts of true polar wander or a dynamo not aligned with the lunar spin axis.' However, the present work (probably correctly) interprets sources in basin antipode zones to be ejecta deposits containing ferromagnetic material from the impactor. These ejecta deposits should have formed almost simultaneously. Even if they had variable thicknesses and cooling rates, they should have acquired their magnetizations within a relatively brief period, geologically speaking. So, how is the directional variability to be explained now? This problem is not limited to the Imbrium antipode zone. Another recent paper has shown that the two main magnetic sources within the Crisium basin are magnetized in different directions (Baek et al., JGR Planets, v. 124, p. 223, 2019). These results appear to be difficult to explain in terms of a centered, axial core dynamo field unless it had an extremely variable orientation. Have the Crisium antipode anomalies been modeled by some of the authors? What were the directional results and what are the implications in view of the present ejecta deposit source interpretation?

(5) The first two authors are from the same institution as the recently deceased H. J. Melosh. The numerical methods section states that the numerical scheme follows that of Melosh et al.

(2017). A number of other references to his work are also given.

He was well aware of the long-standing enigma of lunar magnetism, concentrations antipodal to young impact basins, etc. If he cannot be a co-author, would it be appropriate to mention his contributions to this work (indirect or direct) in the acknowledgments?

Lon Hood

Reviewer #2 (Remarks to the Author):

The manuscript presents a thorough analysis of the possibility that lunar magnetic anomalies are a record of a dynamo field recorded by impact ejecta at impact antipodes, and proposes that their high fields are a result of the ejecta containing impact materials which have higher susceptibilities than those of lunar materials. If true, this work may carry important implications to the study of the origin of lunar magnetism - at present, dynamo models cannot easily explain the large magnetization found in Apollo samples and the field intensities measured from orbit. A higher susceptibility material may resolve this discrepancy, as it requires lower fields to explain the same remanent magnetization. In particular, the authors use detailed impact simulations and detailed analysis of the ejecta trajectories, to show that ejecta of impactor origin may arrive at the antipode before ejecta of lunar origin. This would imply that the high susceptibility material is buried under lunar materials, making the terrain and magnetization consistent with observations.

This work has the potential to complement other studies and add an important piece in the puzzle of lunar magnetism, since as of yet, although the connection between basin antipodes and lunar anomalies was made, the process by which the magnetization would be acquired was not pinned down in detail. The manuscript is therefore of high merit and value to the scientific community.

However, there are a few open questions not addressed by the manuscript that need to be further discussed in order to support the conclusions of the paper.

These include:

1. Linking the anomalies with the basin ejecta: The authors use analysis of the topography of the Crisium antipode, and in particular some craters located near it, to demonstrate that this terrain is likely covered by impact ejecta, for example by showing that those near-by craters are shallower compared to similar-sized craters, implying they were filled after their formation. While this analysis is consistent with the observations, it is not immediately apparent why this would be a leading explanation. While they do discuss the possibility that some of the filling of the crater is due to Orientale impact ejecta, it is not clear why those are the two main options and it would benefit the paper if the authors discussed alternative explanations and why those two impacts are the main options. In particular, the impact simulations and ejecta tracing presented here show that large swaths of the lunar surface can be covered by ejecta from a single event, and that for high impact angle, those ejecta would not be thickest at the antipode. Thus, from a naive perspective, it would appear that many different combinations of impact sites and angles would be able to supply ejecta to a given location on the Moon. Are there other data that would support a specific link between the ejecta and an antipodal basin? Or, alternatively, are there any statistical arguments that can be made that would support the notion that the crater filling by Crisium impact is the likely explanation?

2. The strength of the magnetic anomalies: The authors hypothesize that material of impactor origin would have thermoremanent susceptibility, X_{TRM} , that is a 100 times larger than that of impact melt breccias, implying that the field required to explain the magnetization is therefore a 100 times lower.

First, could the author list a reference about said value of X_{TRM} ?

Second, even with a value that is 100 times larger, this would still imply that a field of 5 uT had to exist on the Moon due to a dynamo at the time of the impact, in order to explain the present-day magnetization.

This magnitude (5 uT) is still larger than that predicted by a convective dynamo model (Evans et al. 2014, 2018). Could the authors discuss this gap and whether they consider another type of dynamo generation mechanism? It would appear, from the information provided, that the extra-lunar material, by itself, still cannot explain “a large fraction of crustal magnetism”, unless some additional assumptions are made.

New references:

Evans, A. J., M. T. Zuber, B. P. Weiss, and S. M. Tikoo (2014), A wet, heterogeneous lunar interior: Lower mantle and core dynamo evolution, *J. Geophys. Res. Planets*, 119, 1061–1077, doi:10.1002/2013JE004494.

Evans, A. J., Tikoo, S. M., & Andrews-Hanna, J. C. (2018). The case against an early lunar dynamo

powered by core convection. *Geophysical Research Letters*, 45, 98–107. <https://doi.org/10.1002/2017GL075441>

3. Impact simulations: Could the authors demonstrate that the impact parameters they chose (100 km impactor and impactor speeds of 12 km/s and 17.4 km/s) are consistent with the formation of a Crisium-sized basin? For this mechanism to be viable, it should be able to both explain the impact basin size and the antipodal arrival times, thickness, and locations. In addition, the paper would benefit from more discussion of how those choices compare to impactor speeds and their resulting basins (e.g., Hood and Artemieva 2008, Miljkovic et al 2013), and the possible variation in the basin formation and ejecta amount with variation in target properties (Miljkovic et al. 2013).

New reference:

Miljkovic et al. (2013) *Science* 08 Nov 2013, Vol. 342, Issue 6159, pp. 724-726, DOI: 10.1126/science.1243224 or

4. Forward modeling to obtain the sub-surface magnetization that explains the anomalies:

In their search for the best-fit distribution of sub-surface dipoles that can produce the observed anomalies, the authors constrain their calculations by the shape of lunar swirls, under the assumption that the strongest magnetization is below those features. While this is a common hypothesis (under the assumption that swirls are caused by blocking of the solar wind), it was not conclusively confirmed. Could the authors provide an estimate of how sensitive their conclusions are to relaxing this assumption? For example, if the swirl contours are not taken into account, would their conclusion change in a substantive way?

5. Analysis of ejecta pattern.

The authors should discuss whether high magnetization is unique to the antipode (or more probable, and by how much) or to low angle impacts.

For the 90 degree impact angle (vertical impact), there is no ejecta reaching the antipode, while at 60 degree impact there is some material reaching the antipode, but only to a very localized area. Further, the authors state in the supplementary material that only impact angles smaller than 45 can supply ejecta to the antipode.

This raises a few questions:

How would those angles compare to the impact angle as deduced from the shape/depth of the impact basin? In other words, can the same impact parameters explain both the impact shape and the ejecta distribution at the antipode?

And finally, if one finds that only a certain impact angle can explain the anomalies under the

mechanism proposed here, how do those compare to the distribution of impact angles of large basin forming impacts? In other words, are these rare occurrences that can cause this magnetization, or would a majority of impacts be able to imprint antipodal magnetization? This point is important for estimating whether this mechanism is the one responsible for much of the magnetization on the Moon, or can only explain very specific anomalies. I suggest the authors add a discussion of this in the main text.

And perhaps in a more general sense, the same high susceptibility material reaches many locations on the Moon, as demonstrated by Fig. 4, and perhaps can lead to magnetization at locations other than the antipode, and for larger impact angles. How plausible would it be that such impactor material, that landed in locations other than antipodes, would explain at least part of the magnetic anomalies ?

Detailed comments:

Main text:

6. Page 2, line 19-20: "This field was likely a dynamo field, instead of an impact-generated field, due to the inefficiencies of recording short-lived fields"

A recent work of relevance here is by Oran et al. (2020), which showed that impact fields alone (without a dynamo) cannot supply the required field magnitude, which strengthen the hypothesis made here in the paper, that the magnetization is indeed a record of a core dynamo.

New reference:

Oran et al. (2020), Was the Moon magnetized by impact plasmas? Science Advances 02 Oct 2020, Vol. 6, no. 40, eabb1475, DOI: 10.1126/sciadv.abb1475

7. Page 4 line 10: ". Thus, oblique impacts are more suitable for producing antipodal ejecta than impacts closer to vertical incidence. As expected, antipodal ejecta arrives later than more proximal ejecta. The earliest antipodal ejecta arrives approximately 10 hours after impact while a small amount of ejecta arrives much later (Supplementary Fig. 6B)."

From inspection of Fig. 6B, it is not clear whether this statement is consistent with the results. First, all ejecta with arrival times between 8 and 24 hours are marked by the same color. Is this what the authors refer to as approximately 10 hours? Second, the ejecta with this coloring are not at the antipode for the 30 degrees impact angle (Fig 6A). The dominant arrival time at the

antipode is between 4-8 hours. For Fig. 6B, which shows the arrival times for a 45 degrees impact, the arrival time at the exact antipode does appear to be between 8-24 hours, but for both impacts, it seems that most of the ejecta arrives much later (24-168 hours) to the vicinity of the antipode, and not only a small fraction.

Could the authors clarify what they consider to be “at the antipode”, and explain why they deduce only a small fraction arrives much later than 10 hours? Second, and more importantly, why is only the 45 degree impact considered in the text and not the 30 degrees (Fig. 6A)? Both are oblique impacts. I suggest that the authors make the quantitative discussion of arrival times, and at what angular distance from the antipode, more precise.

8. Page 5, line 4: How was the value of thermal diffusivity of the ejecta obtained? Please provide a reference.

9. Page 5, line 5: “Although we do not know the size of ejecta fragments “ - could the authors explain why they do not know the size?

10. Page 5, line 7: “We find fragments larger than 0.1–0.5 m are needed to record TRM, when allowing for Curie temperatures for several possible ferromagnetic meteoritic minerals. “

Could you provide more details, here and in the supplement, on how this minimum value was obtained? It would be good if the authors listed the ferromagnetic minerals they consider and their Curie temperatures. More importantly, what is the temperature/arrival time cut off used in order to determine whether a fragment of given temperature and arrival time would still acquire TRM?

11. “The cooling time scale of 10⁻³–10⁻¹ m melt particles is 1s to 2.7 hours, which is shorter than the flight time of antipodal ejecta”

Is there a reference or a calculation to support this?

Figures:

Fig. 2 - could you replace the blue cross with a symbol/cross of another color? As it is, it has the same color as some of the ejecta and it is hard to make out.

The coloring of impact material according to depth: is it possible to draw this with more resolution? For example, now all ejecta from the surface to 20 km deep are marked by the same color. Could you refine this?

Fig. 3: the labeling of the x axis is not clear. Does “flight time” refer to the bottom x axis tick marks, and “cooling timescale of ejecta” to the top x axis? This would appear to be consistent with the text, but it is better to place the axis labels next to their respective axes to avoid ambiguity.

Fig. 4: what is the cause of the arcs of green and blue visible on the right of the antipode? Is

this a numerical artifact? I would presume the distribution of ejecta thickness would be smooth and it is not immediately clear what creates this pattern of interchanging thickness.

Supplementary material

1. Page 2. line 5: "The model is derived from measurements a combination of Lunar Prospector and Kaguya measurements"

The first occurrence of the word "measurements" is probably in error.

2. "we prefer this model to assess correlations with geology"

Since both data types are from the same model, only extracted at different altitudes, I would suggest rephrasing this as:

"so we prefer the model output at 20 km altitude to assess correlations with geology"

3. Page 3: "Note that we take the antipode from the impact site. The impact site and the lunar basin center is the same only for the vertical impact. As the impact site of oblique impact might be different from the lunar basin center, the antipode also does and may have a displacement of a few degrees"

When calculating the antipodes in the various simulations presented here, the impact site is well defined. For data presented, how did you define the impact site? Was it derived from topography of the impact basin, and if so, in what way?

4. Page 3 line 11: "When we derive the thickness on the antipodal hemisphere

" Suggested edit: the thickness -> the ejecta thickness

Reviewer #3 (Remarks to the Author):

Dear Editor and Authors,

This was a very interesting paper that aims to link the origins of intense crustal magnetic

anomalies to antipodally-deposited impact ejecta. The paper used impact simulations to convincingly demonstrate that a substantial amount of ejecta material derived from the target rocks and impactors would be deposited at basin antipodes for a variety of initial impact conditions. If a dynamo field is present at the time of impact, these ejecta deposits will record thermoremanent magnetization (TRM) as they cool below their Curie temperatures. The presence of iron-rich impactor material within these ejecta deposits would lead to acquisition of a very strong TRM compared to regions with only endogenic lunar materials (which contain less metallic iron).

The overall hypothesis has been raised before, but this work does a much more in depth exploration of it than previous works and therefore provides a significant contribution to our understanding of lunar crustal magnetism. The aforementioned conclusions are well justified within the paper and I broadly support publication. However, there are some details that I believe merit further discussion and/or clarification prior to publication:

Broader comments:

This paper brings up anomalies antipodal to the Imbrium, Serenitatis, Orientale, and Crisium basins, but there are other lunar basins (particularly from the farside) that do not appear to have antipodal anomalies such as South Pole-Aitken, Hertzprung, etc. It would be beneficial for the paper to include additional possibilities of why we don't see antipodal anomalies from those. The hypothesis proposed here may be the correct answer, but it still appears a little bit fortuitous that the individual impact parameters from 3 out of 4 basins would lead to ejecta coalescing at this location which just happens to be near a huge basin that would have also produced its own ejecta (and that none of the other impacts save for Orientale would have produced similar antipodal anomalies...).

Some hypotheses for the lack of antipodal anomalies for some basins were advanced in the paper based on impact angle and velocity. However, is it possible that some original impact ejecta material is buried under very thick deposits of ejecta material from later impacts (maybe target rock material which is less magnetic) such that the underlying stronger magnetization is attenuated?

Alternatively, one could say maybe we see southern farside anomalies because the Imbrium and Serenitatis impacts were larger and would have produced more ejecta, but that argument may require addressing Miljkovic et al. (2013) which showed that the nearside basins are larger than they would be otherwise because of heating from the Procellarum KREEP Terrane.

Without addressing the above questions, it is still a tad difficult to disentangle the antipodal ejecta hypothesis from the alternative hypothesis that they represent locally emplaced SP-A ejecta from Wieczorek et al. (2012) as that paper also incorporated an array of impact simulations to justify its conclusions. Maybe both hypotheses contribute?

Smaller comments:

line 34: Oran et al. (2020) demonstrate that the intensity of impact-generated fields alone (i.e., in the absence of a global dynamo field) is very small at the basin antipode and is probably not sufficient to explain strong magnetization on the southern lunar farside. Incorporating the results of that work into the manuscript would be good.

line 43: Tikoo et al. (2015) noted that the intensity of pressure remanent magnetization (PRM), analog for shock remanent magnetization (SRM), is less efficient than thermoremanent magnetization (TRM). This was simply about the remanence mechanism and therefore does not directly mean that recording short-lived fields in an of itself is inefficient. However it may very well be difficult to acquire a substantial TRM from a short-lived field because the timing of cooling below the Curie temperature would have to coincide with the timing of the transient field, and the cooling may be too slow.

line 71: ejecta *deposits*

lines 77 and lines 100-101: The first line says that the earliest ejecta arrives 10 hours after impact while a small amount of ejecta arrives "much later." The latter line says that a 0.5 m fragment has a cooling timescale of 70 hours, which is much less than the flight time of most antipodal ejecta (as 70 > 10). These two statements seem to contradict each other. I tried to look at Supplementary Figure 6 for guidance, but the diagram was confusing to interpret. The caption states "Note that ejecta with longer flight time are *underneath* ejecta with shorter flight times" and this seems very counterintuitive. Shouldn't material with shorter flight times land on the ground first? In one 89 it says "ejecta sourced from the Moon tend to have longer flight times and will bury earlier arriving impactor material" (this makes sense). I think rewriting portions of the text or the figure captions to better explain what is meant would be beneficial.

line 84: Saying impactor material is sourced from "depths of 0-35 km" sounds a bit awkward as "depth" makes it sound like the impactor material is coming from within the Moon. Perhaps say from the outermost 0-35 km of the impactor?

line 185: the statement "This assumption is supported by the fact that swirls are formed by the magnetic field blocking full solar wind access to the surface." is not necessarily correct as other origins such as electromagnetic sorting of the regolith fines or cometary effects are certainly still under consideration. And while blocking of the solar wind may be likely for the main body of Reiner Gamma, other works say that a simple reduced space weathering model for the majority of swirls is unlikely (Pieters 2016 space weathering review paper in JGR). Other parts of this manuscript acknowledge these uncertainties to some degree, but the statement here seems too strong.

line 216: Is that supposed to say 510 microtesla instead of mT? mT units do not match the main text.

lines 218-219: Move the comment that smaller layer thicknesses would produce higher M (an in turn paleofield estimates) up next to lines 206-207. Otherwise while reading, the reader wonders why a 5000 m layer thickness is being used when other parts of the paper demonstrate that thicknesses should be well below 1000 m.

Figure 2 and S5: Why are the color scales for depth within the impactor and target not the same? Is it to qualitatively show that more material comes from the target in the end?

Figure 3 and S7: The paper shows a Curie temperature range spanning from that of pyrrhotite to that of metallic iron, however the inclusion of pyrrhotite is not justified anywhere in the paper or supplement as far as I can tell (?). If this is given because pyrrhotite (formed from aqueous alteration on the parent body) is a primary remanence carrier in carbonaceous chondrites (i.e., impactor material) that should be articulated clearly somewhere. However, given that pyrrhotite has not really been observed in lunar rocks or meteorites, I'm not sure how likely its presence should be given that ejecta experiences melting or thermochemical alteration from heating. The main iron sulfide phase in lunar rocks is troilite.

Responses to the Reviewer #1's Comments:

This manuscript presents a combination of numerical impact simulations and modeling of magnetic anomalies antipodal to the lunar Crisium basin to argue that the observed concentrations of magnetic anomalies antipodal to young large lunar basins can be explained as due to unusually thick concentrations of ejecta containing ferromagnetic material (i.e., iron) from the impactors that created the basins. The numerical simulation results are similar in many ways to those presented previously by Hood and Artemieva (2008), e.g., showing that oblique (less than 45 degrees from horizontal) impacts are needed to produce antipodal ejecta deposition and that the thickness of the deposits is largest at the antipode. The expectation of antipodal concentrations of ejecta goes back at least to ballistic calculations presented by H. J. Moore et al. (Proc. Lunar Sci. Conf. 5th, p. 71, 1974). However, these numerical simulations are more detailed than those reported previously and add new information. For example, the impactor or target origination of the ejecta is shown (Figure 5), allowing an explanation for why the surface composition in basin antipode zones is not more iron-rich than surrounding regions (i.e., ejecta originating in the target arrives later and buries ejecta originating in the impactor). Also, the modeling of anomaly sources antipodal to Crisium shows more clearly that the amplitudes of observed anomalies probably requires stronger magnetization intensities than can be expected for the most ferromagnetic returned lunar samples (i.e., impact melt breccias). This implies that additional ferromagnetic material (probably iron) from the impactor is needed to explain the anomalies.

The issue of whether the observed concentrations of magnetic anomalies antipodal to young large lunar basins implies a physical connection to the basins or is merely coincidental is a long-standing issue of lunar science. Lin et al. (1988; referenced here), using an early large-scale map of lunar crustal fields, presented some simple statistical arguments that the observed strong anomaly concentrations antipodal to four of the youngest large lunar basins (Imbrium, Orientale, Serenitatis, Crisium) are unlikely to have occurred by chance. Mitchell et al. (Icarus, v. 194, p. 401, 2008) presented more detailed Monte Carlo simulations using a better large-scale map to draw a similar conclusion. However, the anomalies antipodal to Imbrium and Serenitatis are also concentrated along the northwestern periphery of the relatively ancient South Pole-Aitken (SPA) basin. This led to the proposal by Wicczorek et al. (2012) that iron from the impactor that created this basin was responsible for the observed anomalies. In fact, the latter reference presented numerical impact simulations and magnetic anomaly modeling peripheral to an impact basin that were similar in many ways to those presented in this manuscript for basin antipode zones.

But evidence for associations of anomalies with young large basins persisted. A later study found a concentration of anomalies near the antipode of the polar Schrödingier basin, which is one of the three youngest lunar basins (Hood et al., JGR Planets, v. 118, doi:10.1002/jgre.20078, 2013). The same study found further evidence for landform modification antipodal to young large basins, which may be consistent with antipodal ejecta deposition. The original proposal that ejecta from young basin-forming impacts are the main sources of orbital magnetic anomalies dates back to the work of D. W. Strangway et al. (Nature Physical Science, v. 246, p. 112, 1973). Most recently, continued mapping of lunar fields has found evidence for elongated anomalies aligned radial to the

Imbrium basin, (Hood, Fall AGU Meeting, 2019). Great circle paths determined by the orientations of the elongated anomalies pass through strong isolated anomalies and intersect in the Imbrium antipode zone where the largest group of strong anomalies is located (Hood et al., under review at JGR Planets at the time of peer-review of the present manuscript – now available at <https://doi.org/10.1029/2020JE006667>).

[redacted]

The current manuscript is therefore a welcome addition to the literature. It builds on the work of Wieczorek et al. (2012) who first showed quantitatively that impactor material (iron) in ejecta from a large basin-forming impact could potentially explain many lunar magnetic anomalies if this material cooled in the presence of an internal magnetizing field. It extends their impact simulations and modeling to the antipodes of the youngest large basins, supporting previous results (e.g., Hood and Artemieva, 2008; Mitchell et al., 2008). However, unlike these latter works, it explains most of the largest concentrations of anomalies as being due to antipodal deposition of ejecta enriched in ferromagnetic material (iron) from the impactors that created the basins. It remains possible that some of the anomalies north of SPA originated at the time of that ancient impact. But the present work shows that a majority of anomalies are more likely to date from the times of the youngest large basin-forming impacts.

Reply: We are grateful for your evaluation of our paper and for your insightful comments, which have helped us to improve the paper. We address each of your comments below and have revised the manuscript accordingly. The major changes are in color highlight text in the revised manuscript.

Based on your nice review of the state of the field, we have added a few references.

I have relatively minor criticisms and questions for the authors to consider in their revision.

(1) The numerical simulations described here assume that both the Moon and the impactor have homogeneous compositions of dunite (mostly olivine). Since the main ferromagnetic carrier in lunar materials is Fe or Fe-Ni, a more realistic approach would be to consider an iron-rich impactor (either with distributed iron or iron in a differentiated core). This was the approach taken by Wieczorek et al. (2012). Can the authors explain briefly in the manuscript why a pure silicate composition is used? The Wieczorek et al. (2012) simulations, for example, tracked the mixing of iron into ejecta and impact melt. The current simulations show that impactor material from different depths in the 100 km diameter impactor ends up as ejecta in the antipodal zone (e.g., Figure 2). This is valuable but there is no explicit determination of how much impactor iron is deposited in the antipodal zone.

Reply: This is an important point. We did not simulate differentiated impactors because the stable release of iSALE-3D cannot handle oblique impacts of differentiated impactors. More technically, iSALE-3D cannot have three materials in a cell (vacuum, impactor mantle, impactor core). We tracked the provenance of impactor material in an attempt to explore how differentiated impactors might affect our results. We now discussed this in the manuscript (Methods).

One reason to choose dunite is to assume the impactor as S-type asteroid, which is similar to ordinary chondrite with 20-30 wt% iron content. We now note this.

“Ordinary chondrites and S-type asteroids, which contain 20-30 wt% iron (Scott and Krot 2014), are well represented by dunite (Svetsov and Shuvalov 2015). Due to the limitation of the current numerical code, we cannot consider differentiated impactors. Since our provenance plot indicates that the outer portion of the impactor dominates the antipodal ejecta (Supplementary Fig. 7), the impactor material can be considered to be mostly mantle material.”

(2) The presented impact simulations show that, for impact angles less than about 45 degrees from the horizontal, concentrated antipodal ejecta deposition occurs (Figures 2 and 4). This agrees with earlier simulations by Natalia Artemieva in Hood and Artemieva (2008). However, the simulations of Wieczorek et al. (2012) found that this range of impact angles also caused all ejecta containing iron from the impactor to be deposited either at the downrange edge of the basin inner rim or outside of the inner rim (see, e.g., their Figures S3A, S3B, and S3C). Crisium has magnetic anomalies well within its inner rim but also concentrated at the antipode. The anomalies within its inner rim suggest a near-vertical impact while the anomalies at the antipode suggest an oblique impact. Please add a few sentences to the text noting this inconsistency and that future simulations tracking the fate of impactor iron are needed to address it. Is it possible that an impact angle close to 45

degrees could produce anomalies both within the inner rim and at the antipode?

Reply: This is a great point. We added this in the discussion.

“Additional high-resolution impact modeling work in the future can also consider more complex ejecta deposition scenarios, such as the emplacement of meteoritic material peripheral to basins, as suggested for Imbrium (Hood et al. 2021), and previously explored for South Pole-Aitken (Wieczorek et al. 2012).”

(3) Unusual modified landforms (referred to by geologists as ‘‘material of grooves and mounds’ and ‘‘hilly and furrowed material’) is found antipodal to young lunar basins, especially Imbrium and Orientale (see lunar geologic maps of these regions dating from the 1970s; for a review, see Hood et al., JGR, 2013). Similar unusual terrain is found antipodal to the Caloris basin on Mercury. Shock effects of converging ejecta impacts are one possible origin. No such terrain is found antipodal to Crisium, possibly because it is overlain by ejecta from the nearby Orientale basin. The existence of such terrain should at least be mentioned because it provides further evidence that antipodal effects of lunar basin-forming impacts are real and are most easily identified for the youngest basins. Is it possible that shock effects of converging ejecta impacts could have contributed to magnetization acquisition in these zones (in addition to TRM)?

Reply: This is a good point. We now note this near the beginning of the paper and cite Hood et al. 2013.

“The antipodes of the Imbrium, Serenitatis, Orientale, and Crisium basins all exhibit large provinces of crustal magnetism and in some cases modified terrain (Hood et al. 2013; Hood and Artemieva 2008; Schultz and Gault 1975).”

We also note the work of Tikoo et al. (2015) who argue that pressure remanent magnetization is difficult to record. We added the following text:

“Pressure remanent magnetization of the rock upon landing is unlikely due to the inefficiencies of this process (Tikoo et al. 2015), the low peak shock pressures (<12 GPa) from ejecta landing at ~2 km/s (Garrick-Bethell et al. 2020), and the randomization of rock-scale remanence directions after initial surface contact.”

(4) Previous work by one of the authors (Nayak, Hemingway, and Garrick-Bethell, *Icarus*, v. 286, p. 153, 2017) found a diverse set of magnetization directions for anomaly sources along the northern edge of SPA (Imbrium antipode zone). The most probable sources in that paper were considered to be igneous intrusions, which could have formed in different magnetic epochs. The directional incoherence was therefore attributed to 'surprisingly large amounts of true polar wander or a dynamo not aligned with the lunar spin axis.' However, the present work (probably correctly) interprets sources in basin antipode zones to be ejecta deposits containing ferromagnetic material from the impactor. These ejecta deposits should have formed almost simultaneously. Even if they had variable thicknesses and cooling rates, they should have acquired their magnetizations within a relatively brief period, geologically speaking. So, how is the directional variability to be explained now?

This problem is not limited to the Imbrium antipode zone. Another recent paper has shown that the two main magnetic sources within the Crisium basin are magnetized in different directions (Baek et al., *JGR Planets*, v. 124, p. 223, 2019). These results appear to be difficult to explain in terms of a centered, axial core dynamo field unless it had an extremely variable orientation. Have the Crisium antipode anomalies been modeled by some of the authors? What were the directional results and what are the implications in view of the present ejecta deposit source interpretation?

Reply: These are good points. Yes, we looked at the magnetization direction in Maxwell and Garrick-Bethell (2019; <https://www.hou.usra.edu/meetings/lpsc2019/pdf/2102.pdf>). The magnetization directions of three of the five identified sub-anomalies are not very different from one another (see their Figure 4). The direction of two of the other anomalies, when analyzed together, is substantially different from the other three. But the two anomalies with the different direction are also the weakest anomalies. Ultimately, a complete analysis with uncertainties is needed to make any strong conclusions about the meaning of these directions (e.g., with the methods in Maxwell and Garrick-Bethell 2019). We also believe that in the future a lunar dynamo that is multi-polar at the surface might become a more realistic possibility, from a modeling perspective.

For now, we added this material to the Methods section:

“We also note that a preliminary analysis of the magnetization directions of several of the strongest anomalies at Crisium’s antipode have similar directions (Maxwell and Garrick-Bethell 2019), which supports the idea they were magnetized in a unidirectional dynamo magnetic field.”

(5) The first two authors are from the same institution as the recently deceased H. J. Melosh. The numerical methods section states that the numerical scheme follows that of Melosh et al.

(2017). A number of other references to his work are also given.

He was well aware of the long-standing enigma of lunar magnetism, concentrations antipodal to young impact basins, etc. If he cannot be a co-author, would it be appropriate to mention his contributions to this work (indirect or direct) in the acknowledgments?

Reply: Thank you for this comment. Although his indirect contributions are difficult to overstate, Jay did not contribute directly to this work. Based on the editor's comment, it seems this is beyond the acknowledgment policy of *Nature Communications*. We leave the acknowledgment as it is now, but will acknowledge him in elsewhere when presenting our work.

Responses to the Reviewer #2's Comments:

The manuscript presents a thorough analysis of the possibility that lunar magnetic anomalies are a record of a dynamo field recorded by impact ejecta at impact antipodes, and proposes that their high fields are a result of the ejecta containing impact materials which have higher susceptibilities than those of lunar materials. If true, this work may carry important implications to the study of the origin of lunar magnetism - at present, dynamo models cannot easily explain the large magnetization found in Apollo samples and the field intensities measured from orbit. A higher susceptibility material may resolve this discrepancy, as it requires lower fields to explain the same remanent magnetization. In particular, the authors use detailed impact simulations and detailed analysis of the ejecta trajectories, to show that ejecta of impactor origin may arrive at the antipode before ejecta of lunar origin. This would imply that the high susceptibility material is buried under lunar materials, making the terrain and magnetization consistent with observations.

This work has the potential to complement other studies and add an important piece in the puzzle of lunar magnetism, since as of yet, although the connection between basin antipodes and lunar anomalies was made, the process by which the magnetization would be acquired was not pinned down in detail. The manuscript is therefore of high merit and value to the scientific community. However, there are a few open questions not addressed by the manuscript that need to be further discussed in order to support the conclusions of the paper.

Reply: We are grateful for your evaluation of our paper and for giving us the detailed comments. We have improved our paper by following your suggestions. The major changes are written in highlight color text in the revised manuscript.

These include:

1. Linking the anomalies with the basin ejecta: The authors use analysis of the topography of the Crisium antipode, and in particular some craters located near it, to demonstrate that this terrain is likely covered by impact ejecta, for example by showing that those near-by craters are shallower compared to similar-sized craters, implying they were filled after their formation. While this analysis is consistent with the observations, it is not immediately apparent why this would be a leading explanation. While they do discuss the possibility that some of the filling of the crater is due to Orientale impact ejecta, it is not clear why those are the two main options and it would benefit the paper if the authors discussed alternative explanations and why those two impacts are the main options. In particular, the impact simulations and ejecta tracing presented here show that large swaths of the lunar surface can be covered by ejecta from a single event, and that for high impact angle, those ejecta would not be thickest at the antipode. Thus, from a naive perspective, it would appear that many different combinations of impact sites and angles would be able to supply ejecta to a given location on the Moon. Are there other data that would support a specific link between the ejecta and an antipodal basin? Or, alternatively, are there any statistical arguments that can be made that would support the notion that the crater filling by Crisium impact is the likely explanation?

Reply: Yes, we do agree that we cannot link the crater fill with Crisium ejecta deposits with 100% certainty. However, if we had *not* found the crater to be partially filled, it would argue against our hypothesis. Hence, this is a necessary but not sufficient condition. A *lack* of fill is certainly a possibility even for old craters: The Nectarian crater Einthoven we analyzed contains almost no fill deposits (Supplementary Fig. 2). Furthermore, the

crater is not just filled, the fill material is *magnetized* (Fig. 1b). We emphasize this is the *only location* we know of on the Moon where the topography (in this case the interior structure of the crater) is correlated with the magnetization pattern. In our opinion this is strong supporting evidence for our hypothesis.

2. The strength of the magnetic anomalies: The authors hypothesize that material of impactor origin would have thermoremanent susceptibility, X_{TRM}, that is a 100 times larger than that of impact melt breccias, implying that the field required to explain the magnetization is therefore a 100 times lower.

First, could the author list a reference about said value of X_{TRM}?

Second, even with a value that is 100 times larger, this would still imply that a field of 5 uT had to exist on the Moon due to a dynamo at the time of the impact, in order to explain the present-day magnetization.

This magnitude (5 uT) is still larger than that predicted by a convective dynamo model (Evans et al. 2014, 2018). Could the authors discuss this gap and whether they consider another type of dynamo generation mechanism? It would appear, from the information provided, that the extra-lunar material, by itself, still cannot explain “a large fraction of crustal magnetism”, unless some additional assumptions are made.

New references:

Evans, A. J., M. T. Zuber, B. P. Weiss, and S. M. Tikoo (2014), A wet, heterogeneous lunar interior: Lower mantle and core dynamo evolution, *J. Geophys. Res. Planets*, 119, 1061–1077, doi:10.1002/2013JE004494.

Evans, A. J., Tikoo, S. M., & Andrews-Hanna, J. C. (2018). The case against an early lunar dynamo powered by core convection. *Geophysical Research Letters*, 45, 98–107. [https://doi.org/ 10.1002/2017GL075441](https://doi.org/10.1002/2017GL075441)

Reply: Answer to first point, we have written the value of X_{TRM} “based on 15 meteorite classes in Table S3 of Wieczorek et al. (2012)” (page 7 in the revised manuscript).

As an answer to second point, our paper must be limited in its scope and unfortunately we do not have sufficient information to assess the type of dynamo. We agree there are still many unknowns in lunar magnetism and how the lunar dynamo was generated, but our work establishing a detailed mechanism to produce at least some of the Moon’s magnetic anomalies is an important step forward. We must remind ourselves that until this work, there has not been an agreed upon geologic formation model for even a single one of the Moon’s magnetic anomalies.

We leave it to the dynamo modeling community to use our work to further our understanding of what type of dynamo may have been active at the time.

3. Impact simulations: Could the authors demonstrate that the impact parameters they chose (100 km impactor and impactor speeds of 12 km/s and 17.4 km/s) are consistent with the formation a Crisium-sized basin? For this mechanism to be viable, it should be able to both explain the impact basin size and the antipodal arrival times, thickness, and locations. In addition, the paper would benefit from more discussion of how those choices compare to impactor speeds and their resulting basins (e.g., Hood and Artemieva 2008, Miljkovic et al 2013), and the possible variation in the basin formation and ejecta amount with variation in target properties (Miljkovic et al. 2013).

New reference:

Miljkovic et al. (2013) *Science* 08 Nov 2013, Vol. 342, Issue 6159, pp. 724-726, DOI: 10.1126/science.1243224 or

Reply: We agree target thermal structure has an effect on resulting basin size. However, since we can apply hydrodynamic scaling to our results, as mentioned in the methods, we do not need to precisely determine the impactor size. We now state the basin size in main text and compared it with previous work in the Supplementary Note:

Main text: **“Note that a 100-km-diameter impact forms a ~1000-km-diameter basin, similar to the diameter of Crisium (Methods, Supplementary Notes).”**

Methods: **“Our 100-km-diameter impactor simulations are valid for roughly 1000-km-diameter scale basins. Using the scaling law (Johnson et al. 2016), a 45° impactor with 100 km in diameter at 12 and 17.4 km/s produces basins that are 930 and 1120 km in diameter, respectively. For comparison Crisium is 1076 km in diameter (Neumann et al. 2015). Assuming hydrodynamic similarity, the ejecta thicknesses reported here can also be scaled linearly by impactor size (Melosh 1989) (see also Supplementary Notes).”**

Supplementary Note: **“We can estimate the impactor size assuming the impact conditions (such as impact velocity, angle, and target properties) using scaling laws. Based on the scaling law (Miljkovic et al. 2016), the Crisium basin (1076 km) can be formed by the impactor of 92 km in diameter with 45° at 12 km/s. Since another scaling law for a different target property (Johnson et al. 2016) also indicates similar results at the same impact velocity and angle (main text), our setting of 100 km is valid for the formation of the Crisium basin. Note that high velocity impact makes a larger basin; e.g., the impactor of 100-200 km at 45° with 18 km/s could form the Imbrium size basin (Hood and Artemieva 2008).”**

4. Forward modeling to obtain the sub-surface magnetization that explains the anomalies: In their search for the best-fit distribution of sub-surface dipoles that can produce the observed anomalies, the authors constrain their calculations by the shape of lunar swirls, under the assumption that the strongest magnetization is below those features. While this is a common hypothesis (under the assumption that swirls are caused by blocking of the solar wind), it was not conclusively confirmed. Could the authors provide an estimate of how sensitive their conclusions are to relaxing this assumption? For example, if the swirls contours are not taken into account, would their conclusion change in a substantive way?

Reply: This is a fair question. Equation 1 in the Methods section shows that the inferred magnetizing field is linearly related to the inferred magnetization M . M is also inversely related to the surface area of the sheet of magnetized material. So, if we let the surface area be larger than the swirls by a factor of two, for example, it would mean the inferred magnetizing field would also be lower by a factor of two. To discuss this effect, we added the sentence in the Methods:

“The field would still be implausibly high (250 μ T) even if the magnetization were spread over an area two times larger than the swirl borders.”

We do not feel it would be reasonable to extend the area of magnetization much more than 2-3 times the swirl borders, since swirls are highly correlated with magnetization. All optically identified swirls are correlated with magnetic anomalies.

5. Analysis of ejecta pattern.

The authors should discuss whether high magnetization is unique to the antipode (or more probable, and by how much) or to low angle impacts.

For the 90 degree impact angle (vertical impact), there is no ejecta reaching the antipode, while at 60 degree impact there is some material reaching the antipode, but only to a very localized area. Further, the authors state in the supplementary material that only impact

angles smaller than 45 can supply ejecta to the antipode.

This raises a few questions:

How would those angles compare to the impact angle as deduced from the shape/depth of the impact basin? In other words, can the same impact parameters explain both the impact shape and the ejecta distribution at the antipode?

And finally, if one finds that only a certain impact angle can explain the anomalies under the mechanism proposed here, how do those compare to the distribution of impact angles of large basin forming impacts? In other words, are these rare occurrences that can cause this magnetization, or would a majority of impacts be able to imprint antipodal magnetization? This point is important for estimating whether this mechanism is the one responsible for much of the magnetization on the Moon, or can only explain very specific anomalies. I suggest the authors add a discussion of this in the main text.

And perhaps in a more general sense, the same high susceptibility material reaches many locations on the Moon, as demonstrated by Fig. 4, and perhaps can lead to magnetization at locations other than the antipode, and for larger impact angles. How plausible would it be that such impactor material, that landed in locations other than antipodes, would explain at least part of the magnetic anomalies?

Reply: Since 45 degree impacts also form circular shape, it is hard to constrain the impact angle from the shape of basins. The original depth of ancient basin is harder to estimate. We noted this in Supplementary Notes.

“Since the shape of basin produced by an impact angle larger than 40° is almost circular (Davison et al. 2011), it is impossible to estimate the impact angle only by its shape. A 30° impact can represent intermediate shape between circular and elliptical. Thus, oblique impacts of 45° and 30° may form nearly circular basins and the antipodal magnetic anomalies at the same time.”

We added the following line for the relationship of the landing site of impactor ejecta and their capability to obtain magnetic fields.

“The diversity in X_{TRM} in meteoritic material, and the diverse differentiation states of impactor bodies, may also explain why some basins form antipodal magnetic anomalies, while others do not.”

Detailed comments:

Main text:

6. Page 2, line 19-20: “This field was likely a dynamo field, instead of an impact-generated field, due to the inefficiencies of recording short-lived fields”

A recent work of relevance here is by Oran et al. (2020), which showed that impact fields alone (without a dynamo) cannot supply the required field magnitude, which strengthens the hypothesis made here in the paper, that the magnetization is indeed a record of a core dynamo.

New reference:

Oran et al. (2020), Was the Moon magnetized by impact plasmas? Science Advances 02 Oct 2020, Vol. 6, no. 40, eabb1475, DOI: 10.1126/sciadv.abb1475

Reply: Thanks, we added this reference to the paper.

7. Page 4 line 10: “. Thus, oblique impacts are more suitable for producing antipodal

ejecta than impacts closer to vertical incidence. As expected, antipodal ejecta arrives later than more proximal ejecta. The earliest antipodal ejecta arrives approximately 10 hours after impact while a small amount of ejecta arrives much later (Supplementary Fig. 6B).” From inspection of Fig. 6B, it is not clear whether this statement is consistent with the results. First, all ejecta with arrival times between 8 and 24 hours are marked by the same color. Is this what the authors refer to as approximately 10 hours? Second, the ejecta with this coloring are not at the antipode for the 30 degrees impact angle (Fig 6A). The dominant arrival time at the antipode is between 4-8 hours. For Fig. 6B, which shows the arrival times for a 45 degrees impact, the arrival time at the exact antipode does appear to be between 8-24 hours, but for both impacts, it seems that most of the ejecta arrives much later (24-168 hours) to the vicinity of the antipode, and not only a small fraction. Could the authors clarify what they consider to be “at the antipode”, and explain why they deduce only a small fraction arrives much later than 10 hours? Second, and more importantly, why is only the 45 degree impact considered in the text and not the 30 degrees (Fig. 6A)? Both are oblique impacts. I suggest that the authors make the quantitative discussion of arrival times, and at what angular distance from the antipode, more precise.

Reply: This was a mistake. We mixed up different impacts which might confuse the reader, so clarified these points in the main text.

“The earliest ejecta from 45° impact landing within 3° of the antipode arrives in approximately 10 hours (Fig. 3; Supplementary Fig. 6) and the flight time of 77% ejecta is within 30 hours. Trajectories of antipodal ejecta from lower impact angles are closer to the Moon’s surface. These shorter flight paths result in smaller flight times (Fig. 3; Supplementary Fig. 6); 93% ejecta produced by the 30° impact land within 3° of the antipode within 10 hours.”

8. Page 5, line 4: How was the value of thermal diffusivity of the ejecta obtained? Please provide a reference.

Reply: We chose a value of 10^{-6} m²/s which is typical thermal diffusivity for rocks. Although we have already referenced Melosh (2011), we moved the reference number ⁽²¹⁾ to the end of the sentence to avoid to be read as a superscript of m.

9. Page 5, line 5: “Although we do not know the size of ejecta fragments “ - could the authors explain why they do not know the size?

Reply: Based on the recent progress of estimating fragmentation (Wiggins et al. 2021) and observed secondary craters (Singer et al. 2020), we deleted this line and rewrote as follows.

“Recent simulations of impact fragmentation suggest that ejecta capable of reaching the antipode produced by a 100-km-diameter impactor will consist of 50-1000 m scale fragments (Wiggins et al. 2021). These estimates are consistent with extrapolation from observation of secondary craters of Orientale (Singer et al. 2020).”

10. Page 5, line 7: “We find fragments larger than 0.1–0.5 m are needed to record TRM, when allowing for Curie temperatures for several possible ferromagnetic meteoritic minerals. “

Could you provide more details, here and in the supplement, on how this minimum value was obtained? It would be good if the authors listed the ferromagnetic minerals they consider and their Curie temperatures. More importantly, what is the temperature/arrival

time cut off used in order to determine whether a fragment of given temperature and arrival time would still acquire TRM?

Reply: We have rephrased to clarify. Essentially the cooling timescale must be longer than the flight time to record TRM. So if the ejecta particles are too small they cannot record TRM because they will already be cold before they are emplaced. In that case, the field they recorded during flight will be weak due to rotation during flight, in addition to being randomized upon landing, and so a strong magnetic anomaly will not form.

“Even ejecta fragments as small as 0.5 m have a cooling timescale, $v_{cool} = 70$ hours, which is longer than the flight time of most antipodal ejecta (> 93%). Thus, for an impact of this scale, we expect most solid ejecta heated to near the Curie temperature would be able to record TRM.”

We have described the Curie temperature of pyrrhotite and iron in the caption of Figure 3.

“Horizontal beige bar shows the corresponding pressure of the Curie temperature of pyrrhotite (593 K) and iron (1043 K)”.

11. “The cooling time scale of 10⁻³–10⁻¹ m melt particles is 1s to 2.7 hours, which is shorter than the flight time of antipodal ejecta”

Is there a reference or a calculation to support this?

Reply: We derived these values from the equation written in the previous paragraph in the main text, $v_{cool} \approx d^2/n$. We now state the times as $v_{cool} = 1s$ to 2.7 hours to clarify this.

Figures:

Fig. 2 - could you replace the blue cross with a symbol/cross of another color? As it is, it has the same color as some of the ejecta and it is hard to make out.

The coloring of impact material according to depth: is it possible to draw this with more resolution? For example, now all ejecta from the surface to 20 km deep are marked by the same color. Could you refine this?

Reply: We changed the blue cross to black cross at the center and also modified the resolution of the depth of the impactor.

Fig. 3: the labeling of the x axis is not clear. Does “flight time” refer to the bottom x axis tick marks, and “cooling timescale of ejecta” to the top x axis? This would appear to be consistent with the text, but it is better to place the axis labels next to their respective axes to avoid ambiguity.

Reply: As we stated above, most ejecta have longer cooling timescale. Thus, we simply deleted the top x axis to avoid confusion.

Fig. 4: what is the cause of the arcs of green and blue visible on the right of the antipode? Is this a numerical artifact? I would presume the distribution of ejecta thickness would be smooth and it is not immediately clear what creates this pattern of interchanging thickness.

Reply: As the reviewer#2 noted, the resolution is one reason to make the arc (see top half circle of Figure 2b). Another is also due to the choice of surface area size (1°), larger area looks smooth (Supplementary Figure 9). We now mentioned this in the captions.

“The arc-shaped patterns on the right side of the antipode appear because of the numerical resolution and the choice of surface area size (Supplementary Notes and Fig. 9).”

Supplementary Notes: **“Although the ejecta thickness changes by the choice of area, our estimates of the ejecta thickness may be conservative and lower limits. This is because secondary craters are distributed heterogeneously (Singer et al. 2020). We**

can expect that the ejecta would also have a heterogeneous distribution (Fig. 4). If we underestimate the ejecta thickness, the field strength is overestimated.”

Supplementary material

1. Page 2. line 5: “The model is derived from measurements a combination of Lunar Prospector and Kaguya measurements”

The first occurrence of the word “measurements” is probably in error.

Reply: We deleted the word.

2. “we prefer this model to assess correlations with geology”

Since both data types are from the same model, only extracted at different altitudes, I would suggest rephrasing this as:

“so we prefer the model output at 20 km altitude to assess correlations with geology”

Reply: We reworded accordingly.

3. Page 3: “Note that we take the antipode from the impact site. The impact site and the lunar basin center is the same only for the vertical impact. As the impact site of oblique impact might be different from the lunar basin center, the antipode also does and may have a displacement of a few degrees”

When calculating the antipodes in the various simulations presented here, the impact site is well defined. For data presented, how did you define the impact site? Was it derived from topography of the impact basin, and if so, in what way?

Reply: We defined the origin of the coordinate as the impact site and the center of simulated basin. However, the impact location of observed lunar basins is poorly constrained. We clarified each basin (simulated or observed) to clarify.

“Note that we take the antipode from the impact site, **which is the origin in the simulation**. The impact site **of the simulation** and the **observed** lunar basin center is the same only for the vertical impact. As the impact site of **simulated** oblique impact might be different from the **observed** lunar basin center, the antipode also does and may have a displacement of a few degrees.”

4. Page 3 line 11: “When we derive the thickness on the antipodal hemisphere “

Suggested edit: the thickness -> the ejecta thickness

Reply: We amended this.

Responses to the Reviewer #3's Comments:

This was a very interesting paper that aims to link the origins of intense crustal magnetic anomalies to antipodally-deposited impact ejecta. The paper used impact simulations to convincingly demonstrate that a substantial amount of ejecta material derived from the target rocks and impactors would be deposited at basin antipodes for a variety of initial impact conditions. If a dynamo field is present at the time of impact, these ejecta deposits will record thermoremanent magnetization (TRM) as they cool below their Curie temperatures. The presence of iron-rich impactor material within these ejecta deposits would lead to acquisition of a very strong TRM compared to regions with only endogenic lunar materials (which contain less metallic iron).

The overall hypothesis has been raised before, but this work does a much more in depth exploration of it than previous works and therefore provides a significant contribution to our understanding of lunar crustal magnetism. The aforementioned conclusions are well justified within the paper and I broadly support publication. However, there are some details that I believe merit further discussion and/or clarification prior to publication:

Reply: We are grateful for your evaluation of our paper and for giving us useful comments. We have improved our manuscript following each of your suggestions below. The major changes are written in highlight text color in the revised manuscript. We believe we have now addressed and resolved each of your comments.

Broader comments:

This paper brings up anomalies antipodal to the Imbrium, Serenitatis, Orientale, and Crisium basins, but there are other lunar basins (particularly from the farside) that do not appear to have antipodal anomalies such as South Pole-Aitken, Hertzprung, etc. It would be beneficial for the paper to include additional possibilities of why we don't see antipodal anomalies from those. The hypothesis proposed here may be the correct answer, but it still appears a little bit fortuitous that the individual impact parameters from 3 out of 4 basins would lead to ejecta coalescing at this location which just happens to be near a huge basin that would have also produced its own ejecta (and that none of the other impacts save for Orientale would have produced similar antipodal anomalies...).

Some hypotheses for the lack of antipodal anomalies for some basins were advanced in the paper based on impact angle and velocity. However, is it possible that some original impact ejecta material is buried under very thick deposits of ejecta material from later impacts (maybe target rock material which is less magnetic) such that the underlying stronger magnetization is attenuated?

Reply: This is an interesting idea. However, the underlying magnetization cannot be easily attenuated unless the overlying material was extremely magnetic itself. For example, most magnetically shielded rooms in paleomagnetism laboratories are made from highly magnetic metals (iron or mu-metal alloy).

Alternatively, one could say maybe we see southern farside anomalies because the Imbrium and Serenitatis impacts were larger and would have produced more ejecta, but that argument may require addressing Miljkovic et al. (2013) which showed that the nearside basins are larger than they would be otherwise because of heating from the Procellarum KREEP Terrane.

Without addressing the above questions, it is still a tad difficult to disentangle the antipodal ejecta hypothesis from the alternative hypothesis that they represent locally emplaced SP-A ejecta from Wieczorek et al. (2012) as that paper also incorporated an array of impact simulations to justify its conclusions. Maybe both hypotheses contribute?

Reply: Yes, we agree there are probably several explanations for the Moon's diverse magnetic anomalies. Our paper is focused on the most detailed explanation to date of just one of those explanations, and demonstrates it is viable by adding important new details through high-resolution 3D simulation.

We suggest that the lack of magnetic anomalies at some basins is most likely function of: 1) The magnetic field strength at the time of the impact, and 2) The magnetic susceptibility of the impactor material. For (1), the field strength is very unconstrained before 3.9 Ga, and also exhibits considerable variability in time subsequently (Weiss and Tikoo 2015).

For (2), the magnetic susceptibility of the impactor material is expected to vary considerably across different impactors. Only a subset of meteoritic material has high X_{TRM} (Wieczorek et al. 2012). The susceptibility of the material will vary with differentiation state and other chemical processes that operated inside the body. For example, Vesta and Ceres each have substantially different thermal histories and compositional properties, and each are large enough such that they would have formed large basins if they had impacted the Moon. A differentiated object like Vesta could have contained substantial amounts of high-susceptibility iron metal in its core. On a partially differentiated object like Ceres, aqueous alteration could have produced substantial magnetite in the body's lower crust, but perhaps much less ferromagnetic material in its rocky core.

To address the reviewer's comments, we have added the following sentence to the section on implications for the lunar magnetic field:

“The diversity in X_{TRM} in meteoritic material, and the diverse differentiation states of impactor bodies, may also explain why some basins form antipodal magnetic anomalies, while others do not.”

We also added this to address the locations other than the antipode.

“Additional high-resolution impact modeling work in the future can also consider more complex ejecta deposition scenarios, such as the emplacement of meteoritic material peripheral to basins, as suggested for Imbrium (Hood et al. 2021), and previously explored for South Pole-Aitken (Wieczorek et al. 2012).”

Smaller comments:

line 34: Oran et al. (2020) demonstrate that the intensity of impact-generated fields alone (i.e., in the absence of a global dynamo field) is very small at the basin antipode and is probably not sufficient to explain strong magnetization on the southern lunar farside. Incorporating the results of that work into the manuscript would be good.

Reply: We now reference Oran et al. (2020) in the introduction.

line 43: Tikoo et al. (2015) noted that the intensity of pressure remanent magnetization (PRM), analog for shock remanent magnetization (SRM), is less efficient than thermoremanent magnetization (TRM). This was simply about the remanence mechanism and therefore does not directly mean that recording short-lived fields in an of itself is inefficient. However it may very well be difficult to acquire a substantial TRM from a short-lived field because the timing of cooling below the Curie temperature would have to coincide with the timing of the transient field, and the cooling may be too slow.

Reply: Yes, we agree. The text was confusing as written. Since Oran et al. 2020 argues strongly against the impact-amplified IMF hypothesis, we have eliminated the text in question. Instead, we added the following text to the beginning of the paper to clarify that we do not expect PRM:

“Pressure remanent magnetization of the rock upon landing is unlikely due to the inefficiencies of this process (Tikoo et al. 2015), the low peak shock pressures (<12 GPa) from ejecta landing at ~2 km/s (Garrick-Bethell et al. 2020), and the randomization of rock-scale remanence directions after initial surface contact.”

line 71: ejecta *deposits*

Reply: We fixed.

lines 77 and lines 100-101: The first line says that the earliest ejecta arrives 10 hours after impact while a small amount of ejecta arrives "much later." The latter line says that a 0.5 m fragment has a cooling timescale of 70 hours, which is much less than the flight time of most antipodal ejecta (as $70 > 10$). These two statements seem to contradict each other. I tried to look at Supplementary Figure 6 for guidance, but the diagram was confusing to interpret. The caption states "Note that ejecta with longer flight time are *underneath* ejecta with shorter flight times" and this seems very counterintuitive. Shouldn't material with shorter flight times land on the ground first? In one 89 it says "ejecta sourced from the Moon tend to have longer flight times and will bury earlier arriving impactor material" (this makes sense). I think rewriting portions of the text or the figure captions to better explain what is meant would be beneficial.

Reply: We thank the reviewer for catching this error! The revised text now correctly reads.

Supplement Figure 6 caption: **“To illustrate the ejecta with short flight times, we plot them on top. Note that in reality, ejecta with shorter flight times will be located underneath ejecta with longer flight times.”**

We also updated main text along with reviewer #2's comment.

Main text: **“These shorter flight paths result in smaller flight times (Fig. 3, Supplementary Fig. 6); 93% ejecta produced by the 30° land within 3° of the antipode within 10 hours.”**

“Even ejecta fragments as small as 0.5 m sized fragment have a cooling timescale, of $\tau_{cool} = 70$ hours, which is longer than the flight time of most antipodal ejecta (> 93%).”

line 84: Saying impactor material is sourced from "depths of 0-35 km" sounds a bit awkward as "depth" makes it sound like the impactor material is coming from within the Moon. Perhaps say from the outermost 0-35 km of the impactor?

Reply: We amended this.

line 185: the statement "This assumption is supported by the fact that swirls are formed by

the magnetic field blocking full solar wind access to the surface." is not necessarily correct as other origins such as electromagnetic sorting of the regolith fines or cometary effects are certainly still under consideration. And while blocking of the solar wind may be likely for the main body of Reiner Gamma, other works say that a simple reduced space weathering model for the majority of swirls is unlikely (Pieters 2016 space weathering review paper in JGR). Other parts of this manuscript acknowledge these uncertainties to some degree, but the statement here seems too strong.

Reply: We agree there isn't 100% certainty and our wording was a bit strong. But even models that invoke unusual regolith properties to explain the brightness (other than space weathering) invoke an interaction between the magnetic field and the solar wind. For example, Garrick-Bethell et al. 2011 invoke electric fields generated by the plasma interaction between the solar wind and magnetic field to perform regolith sorting. Additionally, Hess et al. 2020 argue that the regolith may have anomalously low porosity, and acknowledge that reduced solar wind flux reaching the surface, due to the magnetic field, may be the cause. Even in the cometary model, the magnetization is mostly co-located with the bright soil, and that is all we need to make our argument.

To address the reviewer's concern we softened the statement, removed the wording about weathering, and added two additional references to broaden the possible formation mechanisms:

"This assumes the bright soil at swirls is at least partially the result of the magnetic field blocking full solar wind access to the surface³²⁻³⁶ (added references: Garrick-Bethell et al. 2011; Hess et al. 2020)."

line 216: Is that supposed to say 510 microtesla instead of mT? mT units do not match the main text.

Reply: Thank you for mentioning this. We now fixed to 510 μ T.

lines 218-219: Move the comment that smaller layer thicknesses would produce higher M (an in turn paleofield estimates) up next to lines 206-207. Otherwise while reading, the reader wonders why a 5000 m layer thickness is being used when other parts of the paper demonstrate that thicknesses should be well below 1000 m.

Reply: Following this comment, we moved these lines.

Figure 2 and S5: Why are the color scales for depth within the impactor and target not the same? Is it to qualitatively show that more material comes from the target in the end?

Reply: While the shallower materials (above 25 km) dominate the ejecta sourced from the target, the impactor ejecta originate from the entire body. We modified the color scales of the impactor in Supplementary Figures 2 and 5.

Figure 3 and S7: The paper shows a Curie temperature range spanning from that of pyrrhotite to that of metallic iron, however the inclusion of pyrrhotite is not justified anywhere in the paper or supplement as far as I can tell (?). If this is given because pyrrhotite (formed from aqueous alteration on the parent body) is a primary remanence carrier in carbonaceous chondrites (i.e., impactor material) that should be articulated clearly somewhere. However, given that pyrrhotite has not really been observed in lunar rocks or meteorites, I'm not sure how likely its presence should be given that ejecta experiences melting or thermochemical alteration from heating. The main iron sulfide phase in lunar rocks is troilite.

Reply: We missed mentioning the choice of pyrrhotite and clarified it in the caption. Since

Figure 3 and S7 represents the ejecta from the impactor, pyrrhotite can come from the impactor, not the lunar materials.

“We take pyrrhotite as a lower limit of Curie Temperature. Pyrrhotite is found in carbonaceous chondrite, possible impactors, as a remanent carrier (Weiss et al. 2021).”

REVIEWER COMMENTS

Reviewer #2 (Remarks to the Author):

The authors made thoughtful and thorough changes in the manuscript and addressed most points raised by Reviewer #1 and Reviewer #2 satisfactorily.

There are still a few points that need to be clarified that pertain to the main conclusions of the paper and that merit clarification.

1. General comment:

The introduction seems to end quite abruptly; the authors suggest that impactor materials with high susceptibility may be able to explain the discrepancy between the strong magnetization of crustal materials and magnetic fields predicted by dynamo theory. From there, the paper goes straight to the results and describe the distribution of the antipodal ejecta. There is no mention of the paper would text this hypothesis by running impact simulations, tracing the distribution of ejecta, and estimating the resulting magnetization. While the technical details of the simulations are in the supplement, the main text should at least mention what the paper aims to do, and define what specific gaps in the existing body of similar modeling they aim to fill.

2. Motivation and comparison to predictions of dynamo theory:

The paper is lacking in citation and quoting numerical values of the dipole field that is hypothesized to magnetize the ejecta. See also previous comment #2 from Reviewer #2. It is agreed that an in-depth exploration of dynamo theory is not in the scope of the paper. However, since the main hypothesis tested by the paper is that meteoritic ejecta can explain the magnetization, there should be at least minimal discussion of gap between the inferred magnetizing field and the range of surface fields predicted by lunar dynamos. Now, the only mention is in line 66 of the revised text, stating that an ancient field of 500 uT is "hard to explain". An initiated reader cannot estimate how far-reaching a 500 uT field it, and why a 100 times larger susceptibility would "do the trick". In fact, this can strengthen the conclusion of the paper, since a X_{TRM} that is a factor 100 larger would basically mean that a 5 uT field is sufficient. Showing that a 5 uT field is not far from dynamo theory predictions can very nicely tie this all together.

As an aside, an ancient magnetizing field of 500 uT derived here from forward modeling of lunar swirls is much larger than magnetizing fields derived by analysis of returned samples (Weiss and Tikoo 2014, Wieczorek et al. 2012), which only implies tens of uT as the likely highest ancient fields. Could you address this gap?

3. Crater size:

In response to the previous comments by Reviewer #2, the authors now report the size of the crater that their simulation would produce, given the chosen impact parameters - a basin with a diameter of 1000 km. This is indeed consistent with the size of Crisium. However, the authors mention this was calculated using a scaling law (without providing the details), from which it could be concluded that this is not a result of the simulation.

While a scaling law may be satisfactory in some contexts, the original comment was more subtle: because the simulations are aimed at showing that the simulated ejecta can explain the magnetization at the antipode, and because using different assumptions in impact simulations may produce not only a different basin but also a different ejecta pattern, arrival times, and blanket thickness, it is important to show that the assumptions used, including the composition and thermal structure of the crust and mantle, is the appropriate one, and that the results do indeed support the hypothesis that this is magnetization formed by, or consistent with, the Crisium event.

In this context, it is important to note the work by Potter et al. (2012). This work used a revised thermal profile and demonstrated it significantly affects the resulting basin size, excavated volume, excavated depth, and other parameters. It is important to note they used the same numerical code (iSALE) as was used here so comparing the assumptions and results is pertinent. In particular, Table 2 and Table 3 in Potter et al. show that an impactor with a 100 km diameter hitting the Moon at 10 km/s creates a much smaller crater, of a diameter of 500 km, and not 1000 km obtained from the scaling law in the present manuscript. More so, Hood & Artemieva (2008) simulated the formation of the same four young large basins considered here using an impactor of 240 km in diameter with a speed of 18 km/s. Using revised thermal profiles, Potter et al. showed that a 240 km impactors would form an SPA type basin, as seen in Table 2 therein. This suggests that impactors associated with a specific basin may be in fact smaller than previously assumed.

In summary, it is advised that the authors discuss:

- Why is a scaling law used rather than the simulated basin? Is this due to a computational limitation of running the simulation until the basin settles? If so, please state this.
- Compare their work and choice of impact parameters to Potter et al.
- If indeed a smaller impactor may be more appropriate, as suggested by the above, please explain whether the resulting ejecta distribution, travel time, or thickness, are expected to change. Specifically - would they change in a way that would alter the conclusions? Would this uncertainty affect the likelihood of antipodal magnetization being able to explain lunar crustal magnetization?

While I agree with the authors' rebuttal that there are many uncertainties in lunar magnetism that are yet to be explored, and that this paper makes an important step forward, it is also important to state those uncertainties and their possible implications, such that previous and future works could be effectively compared.

4. Another point about the impact properties was raised in Comment #2 of Reviewer #1, pointing to the discrepancy between the oblique impacts considered here, which are needed to bring ejecta to the antipode, and near-vertical impacts needed to explain the fact that Crisium has magnetic anomalies within its inner rim. This discrepancy needs to be explicitly addressed, since, as in item 3 above, the same simulation should both create the basin and the antipodal formations. In response, the authors modified the text to state that future work "can consider more complex ejecta deposition scenarios". This statement is too vague, and more importantly, does not address the discrepancy pointed to by the review.

Minor comments:

1. In comment #1 from Reviewer #2, the authors were asked whether impacts other than the Crisium-forming event could have filled the craters near the antipode. While this reviewer agrees that there is no unequivocal method to link crater fill with a specific impact event, and that the evidence included here is convincing that the basins are indeed filled by ejecta, it is this same difficulty in linking fill with an impact that plays a role in the difficulty in linking a specific magnetization with an antipode. This is mainly because impact simulations show that ejecta can cover huge areas. It is quite true that this work presents the most advanced effort to date in linking antipodes and magnetization, but the inherent uncertainties should be mentioned in a sentence or two.

2. In line 70, it is stated that meteoritic materials can have X_{TRM} up to 100 higher than lunar material. Could you add a citation for this? While this is later discussed again in the discussion, a reference here would be useful.

3. In Line 147, the authors state as a conclusion of their work: "The impactor material found at basin antipodes can explain a substantial fraction of the Moon's crustal magnetism". This statement may be too strong. This work makes a very important step forward by explicitly demonstrating that impact ejecta patterns are consistent with the magnetization, and offers a unique contribution by illustrating the key role of impactor materials in making the antipodal magnetization hypothesis

viable. The covering of such materials by late-arriving ejecta of lunar origin is also a strong and elegant conclusion. At the same, the paper did not discuss whether this mechanism can explain a significant part of all magnetization found on the Moon. It is perhaps better to say that it can explain a large portion of the range of magnetization found on the Moon. Further, it is not clear whether the magnetization is unique to the antipodes, or perhaps can explain magnetization in other locations far from the basin. This is an important point, as the implications of mostly antipodal magnetization vs. a more spread out pattern would affect the trajectory of how this problem needs to be further studied.

Reviewer #3 (Remarks to the Author):

One of the most enigmatic aspects of lunar magnetism is the origin of the strongest crustal magnetic anomalies on the Moon. Many of these anomalies happen to be co-located with the antipodes of large impact basins.

In their revised manuscript, using 3-D iSALE impact simulations, Wakita et al. present a compelling argument that a substantial amount of impact ejecta (>hundreds of meters thickness, comprised of material from both the impactor and the target) are able to accumulate at basin antipodes. Because meteoritic material is more iron rich than lunar crustal rocks, the addition of exogenic iron can contribute to produce the observed intensity of magnetic anomalies at basin antipodes. The presence of lunar swirls at many basin antipodes and the intensities of their magnetizations suggest the relevant magnetic source bodies be shallow and have narrow lateral length scales, which can be consistent with individual impact ejecta deposits.

Overall, I feel the revised version of this paper has addressed most of the previous round of review comments satisfactorily and has made some compelling arguments in favor of an impactor material contribution to antipodal anomalies. As such, this paper will be a welcome addition to the lunar magnetism literature. I do have a few lingering comments that would be good to address, if possible, but I don't necessarily consider them a barrier to publication:

1. Lines 102-103. The equation used to estimate the cooling timescale is (if I correctly recall) for conductive cooling so I think these values might need to be presented as upper limits since they don't consider convection and radiation. Also this expression doesn't get into what high temperature to what low temperature this timescale relates to with respect to the magnetism acquisition temperature range. Related: I think in Figure 3 the Curie temperatures shown (with respect to the pressure) are being compared to the peak temperature associated with a given peak pressure (maybe clarify this in the caption...). The temperature that these ejecta materials are being emplaced are likely to be lower, as mentioned in the text so without some additional info (see comment 2 below) it is hard to see how this info translates into the amount of magnetizable material post-emplacment.

2. Lines 111-113. I think it would be good, if possible, to estimate (based on the peak pressures and temperatures experienced by the ejecta in Figure 3) the amount of impactor sourced ejecta (in particular) that would undergo melting and then experience rapid droplet cooling prior to emplacement on the surface. This would tell us something about what fraction of the antipodal ejecta could actually contribute to the remanence at basin antipodes.

2. Could a sentence be added about to what extent different planetesimal sizes may be expected to be differentiated based on planetesimal evolution modeling (Hevey and Sanders, 2006 or Bryson et al., 2019)? That would be useful for contextualizing how magnetic one might expect the outermost 35 km of a 100 km planetesimal should be. Of course, there are many uncertainties associated with impactor composition (as mentioned in lines 140-142) but an additional sentence there would be nice.

3. I think there may be enough information in these models to explore (based on the amount of ejecta sourced from the impactor that lands on places on the Moon that are not antipodal to impact sites, as documented in the paper's figures) whether other strong magnetic anomalies and lunar swirl source bodies could be sourced from impactors (as suggested by Hood et al. recently). The authors suggest that this could be explored in future work (lines 152-155) but I'm not sure how the simulations would be conducted any differently than what was done for this study.

4. In line 246, I don't follow how 1-km resolution translates to 50 cells per projectile for a 100-km diameter impactor. Could some more information be added to better understand how the iSALE model is set up with respect to that resolution?

Suggested small text corrections:

Line 126: the Crisium *antipode*'s strongest anomalies

Line 251-252: The outer portion of the impactor may only be considered to be "mantle" material for a differentiated impactor that has a silicate mantle. While some 100 km diameter bodies are differentiated in this way, some may not be. May want to broaden how this is phrased.

Figure 2: the impactor angle is incorrectly given in part d and says 30 degrees instead of 90 degrees. There is at least one figure in the supplement (possibly more...) where the impactor angles in the figure and caption do not appear match as well.

Line 444: *remanence* carrier

Responses to the Reviewer #2's Comments:

The authors made thoughtful and thorough changes in the manuscript and addressed most points raised by Reviewer #1 and Reviewer #2 satisfactorily.

There are still a few points that need to be clarified that pertain to the main conclusions of the paper and that merit clarification.

1. General comment:

The introduction seems to end quite abruptly; the authors suggest that impactor materials with high susceptibility may be able to explain the discrepancy between the strong magnetization of crustal materials and magnetic fields predicted by dynamo theory. From there, the paper goes straight to the results and describe the distribution of the antipodal ejecta. There is no mention of the paper would text this hypothesis by running impact simulations, tracing the distribution of ejecta, and estimating the resulting magnetization. While the technical details of the simulations are in the supplement, the main text should at least mention what the paper aims to do, and define what specific gaps in the existing body of similar modeling they aim to fill.

Reply: We thank the reviewer for pointing this out. We added this at the end of the Introduction.

“Using high-resolution impact simulations we explore the hypothesis that antipodal ejecta contains sufficient impactor material to explain the observed magnetization of anomalies antipodal to large basins. Previous exploration of antipodal ejecta deposits did not explore the fate of impactor materials (Hood and Artemieva 2008). For moderately oblique impacts we find that antipodal ejecta is dominated by the impact materials which can have high X_{TRM} like the chondritic meteorite. Moreover, this ejecta is above the Curie temperature at the time of emplacement and can thus record the magnetic field of the Moon as it cools.”

2. Motivation and comparison to predictions of dynamo theory:

The paper is lacking in citation and quoting numerical values of the dipole field that is hypothesized to magnetize the ejecta. See also previous comment #2 from Reviewer #2. It is agreed that an in-depth exploration of dynamo theory is not in the scope of the paper. However, since the main hypothesis tested by the paper is that meteoritic ejecta can explain the magnetization, there should be at least minimal discussion of gap between the inferred magnetizing field and the range of surface fields predicted by lunar dynamos. Now, the only mention is in line 66 of the revised text, stating that an ancient field of 500 uT is “hard to explain”. An initiated reader cannot estimate how far-reaching a 500 uT field it, and why a 100 times larger susceptibility would "do the trick". In fact, this can strengthen the conclusion of the paper, since a X_{TRM} that is a factor 100 larger would basically mean that a 5 uT field is sufficient. Showing that a 5 uT field is not far from dynamo theory predictions can very nicely tie this all together.

Reply: We thank the reviewer for this suggestion and have added additional text (see below) to the relevant section that clarifies that a 500-uT-field is unrealistically large.

As an aside, an ancient magnetizing field of 500 uT derived here from forward modeling of lunar swirls is much larger than magnetizing fields derived by analysis of returned samples (Weiss and

Tikoo 2014, Wieczorek et al. 2012), which only implies tens of uT as the likely highest ancient fields. Could you address this gap?

We added a sentence to the same paragraph as the material above, explaining that 500 uT fields are also too high relative to the values inferred from sample analysis.

“Such high inferred dynamo field strengths of 500 μT are likely untenable, since current theory suggests that lunar paleofields up to only $\sim 3\text{-}15 \mu\text{T}$ may be achievable (Weiss and Tikoo et al. 2014; Evans et al. 2018). Even the paleofield values inferred from most strongly magnetized Apollo samples are $\sim 5\text{-}10$ times weaker than 500 μT (Weiss and Tikoo 2014). Because the inferred paleofield strength is inversely proportional to assumed X_{TRM} , our forward model implies material at the Crisium antipode with a $X_{\text{TRM}} \sim 30\text{-}150$ times higher than lunar materials would be consistent with the $\sim 3\text{-}15 \mu\text{T}$ dynamo field limits predicted by theory.”

3. Crater size:

In response to the previous comments by Reviewer #2, the authors now report the size of the crater that their simulation would produce, given the chosen impact parameters - a basin with a diameter of 1000 km. This is indeed consistent with the size of Crisium. However, the authors mention this was calculated using a scaling law (without providing the details), from which it could be concluded that this is not a result of the simulation.

While a scaling law may be satisfactory in some contexts, the original comment was more subtle: because the simulations are aimed at showing that the simulated ejecta can explain the magnetization at the antipode, and because using different assumptions in impact simulations may produce not only a different basin but also a different ejecta pattern, arrival times, and blanket thickness, it is important to show that the assumptions used, including the composition and thermal structure of the crust and mantle, is the appropriate one, and that the results do indeed support the hypothesis that this is magnetization formed by, or consistent with, the Crisium event.

In this context, it is important to note the work by Potter et al. (2012). This work used a revised thermal profile and demonstrated it significantly affects the resulting basin size, excavated volume, excavated depth, and other parameters. It is important to note the used the same numerical code (iSALE) as was used here so comparing the assumptions and results is pertinent. In particular, Table 2 and Table 3 in Potter et al. show that an impactor with a 100 km diameter hitting the Moon at 10 km/s creates a much smaller crater, of a diameter of 500 km, and not 1000 km obtained from the scaling law in the present manuscript. More so, Hood & Artemieva (2008) simulated the formation of the same four young large basins considered here using an impactor of 240 km in diameter with a speed of 18 km/s. Using revised thermal profiles, Potter et al. showed that a 240 km impactors would form an SPA type basin, as seen in Table 2 therein. This suggests that impactors associated with a specific basin may be in fact smaller than previously assumed.

In summary, it is advised that the authors discuss:

- Why is a scaling law used rather than the simulated basin? Is this due to a computational limitation of running the simulation until the basin settles? If so, please state this.
- Compare their work and choice of impact parameters to Potter et al.

- If indeed a smaller impactor may be more appropriate, as suggested by the above, please explain whether the resulting ejecta distribution, travel time, or thickness, are expected to change. Specifically - would they change in a way that would alter the conclusions? Would this uncertainty affect the likelihood of antipodal magnetization being able to explain lunar crustal magnetization?

While I agree with the authors' rebuttal that there are many uncertainties in lunar magnetism that are yet to be explored, and that this paper makes an important step forward, it is also important to state those uncertainties and their possible implications, such that previous and future works could be effectively compared.

Reply: We have performed an additional simulation to ensure our impactor size is reasonable. Our current work focused on the ejecta from the crater forming impacts which requires high resolution (1 km) within a short timescale (< 100 s). Later time ejecta would have lower velocity and could not reach the antipode. To evaluate the final size of crater, longer timescale (> 1 hours) is necessary and it is computationally expensive with our current resolution. Thus, we have performed one simulation with a lower resolution to ensure our impactor is a reasonable size (see next response and Supplementary Notes, Figs. 10, and 11).

Note also the crater size reported in Potter et al. (2012) is the transient crater diameter, which can be given at the maximum excavation volume during impact. This is not the final crater size as we observed on Moon. The final crater diameter is about 1.5-2 times larger than the transient one (e.g., Johnson et al. 2016). For example, Miljković et al. (2016) reported the transient crater of the Crisium basin is 389 km. Our lower resolution run (10 km resolution) produces a transient crater ~400 km in diameter. Because this is consistent with the transient crater size Miljković et al. (2016) found was needed to reproduce Crisium, we are confident our impactor size is the correct size for producing a Crisium scale basin. Although it is hard to determine precisely, the final crater is about the size of Crisium.

Please see the extensive additions to the supplementary material discussing this.

4. Another point about the impact properties was raised in Comment #2 of Reviewer #1, pointing to the discrepancy between the oblique impacts considered here, which are needed to bring ejecta to the antipode, and near-vertical impacts needed to explain the fact that Crisium has magnetic anomalies within its inner rim. This discrepancy needs to be explicitly addressed, since, as in item 3 above, the same simulation should both create the basin and the antipodal formations. In response, the authors modified the text to state that future work "can consider more complex ejecta deposition scenarios". This statement is too vague, and more importantly, does not address the discrepancy pointed to by the review.

Reply: Additional to the previous response, we have performed low-resolution run (Supplementary Notes, Supplementary Figs. 10 and 11) to explore the distribution of impactor material in and around the basin. We noted this in the main text.

"We ran our simulation of a 100-km-diameter impactor with 45° at 12 km/s at lower resolution until the final crater has formed. This simulation indicates that, in addition to producing antipodal anomalies, shock-heated impactor materials will also be distributed in and around large basins (Supplementary Notes, Supplementary Figs. 10, and 11). These

results indicate that an impactor material from a basin forming impact can produce antipodal anomalies as well as the magnetic anomalies found in and around large basins (e.g., Imbrium (Hood et al. 2021), South Pole-Aitken basin (Wieczorek et al. 2012), and Crisium (Baek et al. 2019)).”

Minor comments:

1. In comment #1 from Reviewer #2, the authors were asked whether impacts other than the Crisium-forming event could have filled the craters near the antipode. While this reviewer agrees that there is no unequivocal method to link crater fill with a specific impact event, and that the evidence included here is convincing that the basins are indeed filled by ejecta, it is this same difficulty in linking fill with an impact that plays a role in the difficulty in linking a specific magnetization with an antipode. This is mainly because impact simulations show that ejecta can cover huge areas. It is quite true that this work presents the most advanced effort to date in linking antipodes and magnetization, but the inherent uncertainties should be mentioned in a sentence or two.

Reply: We added the following in Methods.

“Although the ejecta in our simulations cover a large area and the variation of ejecta are relatively smooth (Fig. 4), distal ejecta deposition is an inherently stochastic process, often complicated by heterogeneous contributions from rays (Speyerer et al. 2016), and we acknowledge that it cannot be modeled completely accurately with current simulations.”

2. In line 70, it is stated that meteoritic materials can have X_{TRM} up to 100 higher than lunar material. Could you add a citation for this? While this is later discussed again in the discussion, a reference here would be useful.

Reply: We now referenced Wieczorek et al. (2012).

3. In Line 147, the authors state as a conclusion of their work: “The impactor material found at basin antipodes can explain a substantial fraction of the Moon’s crustal magnetism”. This statement may be too strong. This work makes a very important step forward by explicitly demonstrating that impact ejecta patterns are consistent with the magnetization, and offers a unique contribution by illustrating the key role of impactor materials in making the antipodal magnetization hypothesis viable. The covering of such materials by late-arriving ejecta of lunar origin is also a strong and elegant conclusion. At the same, the paper did not discuss whether this mechanism can explain a significant part of all magnetization found on the Moon. It is perhaps better to say that it can explain a large portion of the range of magnetization found on the Moon. Further, it is not clear whether the magnetization is unique to the antipodes, or perhaps can explain magnetization in other locations far from the basin. This is an important point, as the implications of mostly antipodal magnetization vs. a more spread out pattern would affect the trajectory of how this problem needs to be further studied.

Reply: We reworded this.

“The impactor material found at basin antipodes can explain some of the Moon’s enigmatic magnetic anomalies.”

Responses to the Reviewer #3's Comments:

One of the most enigmatic aspects of lunar magnetism is the origin of the strongest crustal magnetic anomalies on the Moon. Many of these anomalies happen to be co-located with the antipodes of large impact basins.

In their revised manuscript, using 3-D iSALE impact simulations, Wakita et al. present a compelling argument that a substantial amount of impact ejecta (>hundreds of meters thickness, comprised of material from both the impactor and the target) are able to accumulate at basin antipodes. Because meteoritic material is more iron rich than lunar crustal rocks, the addition of exogenic iron can contribute to produce the observed intensity of magnetic anomalies at basin antipodes. The presence of lunar swirls at many basin antipodes and the intensities of their magnetizations suggest the relevant magnetic source bodies be shallow and have narrow lateral length scales, which can be consistent with individual impact ejecta deposits.

Overall, I feel the revised version of this paper has addressed most of the previous round of review comments satisfactorily and has made some compelling arguments in favor of an impactor material contribution to antipodal anomalies. As such, this paper will be a welcome addition to the lunar magnetism literature. I do have a few lingering comments that would be good to address, if possible, but I don't necessarily consider them a barrier to publication:

Reply: We are very grateful for your evaluation of our paper and for your helpful comments. We have revised the manuscript accordingly. The major changes are in color highlight text in the revised manuscript.

1. Lines 102-103. The equation used to estimate the cooling timescale is (if I correctly recall) for conductive cooling so I think these values might need to be presented as upper limits since they don't consider convection and radiation. Also this expression doesn't get into what high temperature to what low temperature this timescale relates to with respect to the magnetism acquisition temperature range. Related: I think in Figure 3 the Curie temperatures shown (with respect to the pressure) are being compared to the peak temperature associated with a given peak pressure (maybe clarify this in the caption...). The temperature that these ejecta materials are being emplaced are likely to be lower, as mentioned in the text so without some additional info (see comment 2 below) it is hard to see how this info translates into the amount of magnetizable material post-emplacment.

Reply: This is actually a lower limit on the cooling timescale. The cooling ultimately happens from radiation from the surface of an ejecta fragment, but the heat must conduct to the surface. Conduction is the limiting timescale in this process (e.g., Melosh, 2013 Planetary surface processes).

In addition, including radiation from nearby particles will actually increase the timescale of cooling as we indicate. Convection could certainly speed up the cooling timescale, but since we are focused on solid fragments, convection should not occur.

We have updated the text to clarify.

“The cooling of fragments is limited by the rate of conduction and the cooling timescale of ejecta is $\tau_{\text{cool}} \approx d^2/\kappa$, where d is their size and $\kappa = 10^{-6} \text{ m}^2/\text{s}$ is the thermal diffusivity [22].”

2. Lines 111-113. I think it would be good, if possible, to estimate (based on the peak pressures and temperatures experienced by the ejecta in Figure 3) the amount of impactor sourced ejecta (in particular) that would undergo melting and then experience rapid droplet cooling prior to emplacement on the surface. This would tell us something about what fraction of the antipodal ejecta could actually contribute to the remanence at basin antipodes.

Reply: Good idea. The melt fraction is minor for the 12 km/s cases but not for 17.4 km/s. We added following to mention this point.

“... (Supplementary Fig. 8); 89% and 15% of antipodal ejecta from the 45° impacts is unmelted for impact velocities of 12 and 17.4 km/s, respectively.”

2. Could a sentence be added about to what extent different planetesimal sizes may be expected to be differentiated based on planetesimal evolution modeling (Hevey and Sanders, 2006 or Bryson et al., 2019)? That would be useful for contextualizing how magnetic one might expect the outermost 35 km of a 100 km planetesimal should be. Of course, there are many uncertainties associated with impactor composition (as mentioned in lines 140-142) but an additional sentence there would be nice.

Reply: The differentiation states depend on the formation time of planetesimals rather than its size. We briefly discussed this as follows and also noted this in the Methods (see below).

“For example, early accreting impactors larger than ~40 km in diameter tend to be differentiated while those that accreted later may remain undifferentiated (Hevey and Sanders, 2006; Gail et al. 2014, see Methods).”

3. I think there may be enough information in these models to explore (based on the amount of ejecta sourced from the impactor that lands on places on the Moon that are not antipodal to impact sites, as documented in the paper's figures) whether other strong magnetic anomalies and lunar swirl source bodies could be sourced from impactors (as suggested by Hood et al. recently). The authors suggest that this could be explored in future work (lines 152-155) but I'm not sure how the simulations would be conducted any differently than what was done for this study.

Reply: This is a good idea. Our current work focused on the ejecta from the crater-forming impacts, which requires reasonable time with 1 km resolution to simulate. The crater formation with the same resolution is computationally expensive. Thus, we have performed one test run with low resolution to check this and find that the same impact that produces a magnetic anomaly will also distribute impactor material in and around the basin consistent with observed anomalies.

“We ran our simulation of a 100-km-diameter impactor with 45° at 12 km/s at lower resolution until the final crater has formed. This simulation indicates that, in addition to producing antipodal anomalies, shock-heated impactor materials will also be distributed in and around large basins (Supplementary Notes, Supplementary Figs. 10, and 11). These results indicate that an impactor material from a basin forming impact can produce antipodal anomalies as well as the magnetic anomalies found in and around large basins (e.g., Imbrium (Hood et al. 2021), South Pole-Aitken basin (Wieczorek et al. 2012), and Crisium (Baek et al. 2019)).”

4. In line 246, I don't follow how 1-km resolution translates to 50 cells per projectile for a 100-km diameter impactor. Could some more information be added to better understand how the iSALE model is set up with respect to that resolution?

Reply: We now clarified the cells size as 50 cells per projectile ***radius***.

Suggested small text corrections:

Line 126: the Crisium ***antipode***'s strongest anomalies

Reply: We amended this.

Line 251-252: The outer portion of the impactor may only be considered to be "mantle" material for a differentiated impactor that has a silicate mantle. While some 100 km diameter bodies are differentiated in this way, some may not be. May want to broaden how this is phrased.

Reply: We now note this in the Methods.

“While the impactor formed within 1-2 Myr after the birth of the Solar System could be fully differentiated like iron meteorites (Kruijer et al. 2013), the later formed bodies could be partially differentiated or remain undifferentiated like ordinary chondrites (Bryson et al. 2019, Blackburn et al. 2017, Gail et al. 2014). Since our provenance plot indicates that the outer portion of the impactor dominates the antipodal ejecta (Supplementary Fig. 7), the antipodal ejecta can be considered to be mantle or undifferentiated material.”

Figure 2: the impactor angle is incorrectly given in part d and says 30 degrees instead of 90 degrees. There is at least one figure in the supplement (possibly more...) where the impactor angles in the figure and caption do not appear match as well.

Reply: We really thank the reviewer for catching this error! We fixed the text label in Figure 2d and Supplementary Figure 5d.

Line 444: ***remanence*** carrier

Reply: We amended this.

REVIEWER COMMENTS

Reviewer #2 (Remarks to the Author):

The authors have addressed all outstanding issues from the previous review satisfactorily, and added simulations, plots, and text to support their findings. As such, I highly recommend the publication of this work.

There is one location where new added text needs to be more carefully stated. The line in question is (line 274), which states:

"While the impactor formed within 1-2 Myr after the birth of the Solar System could be fully differentiated like iron meteorites, the later formed bodies could be partially differentiated or remain undifferentiated like ordinary chondrites"

This needs to be stated more carefully. First, a distinction has to be made between meteorites and their parent bodies. Iron meteorites, or chondritic meteorites are not by themselves differentiated or not, but rather it is their parent bodies that may be so. Second, both may samples of different layer within a partially differentiated bodies (Elkins-Tanton et al. 2011,) and as of yet, there is no conclusive answer to whether or not chondrites come from undifferentiated parent bodies or whether iron meteorites come from fully differentiation parent bodies. In fact, several lines of evidence suggest partial differentiated asteroids may be the parent bodies of chondrites (Carporzen et al. 2011, Oran et al. 2018, Bryson et al. 2019) and well as some iron meteorites (Maurel et al. 2020). In general, the identification of meteorite parent bodies and their histories are still vigorously investigated (Vernazza et al. 2015, Greenwood et al. 2020)

I suggest editing this sentence, to both more accurately present the current state of knowledge of meteorite parent bodies, and also to clarify the context of this new text - and in particular how does the age and state of differentiation of possible impactors fits into the general hypothesis being tested here.

New references:

Carporzen, Laurent, et al. "Magnetic evidence for a partially differentiated carbonaceous chondrite parent body." *Proceedings of the National Academy of Sciences* 108.16 (2011): 6386-6389.

Elkins-Tanton, Linda T., Benjamin P. Weiss, and Maria T. Zuber. "Chondrites as samples of differentiated planetesimals." *Earth and Planetary Science Letters* 305.1-2 (2011): 1-10.

Greenwood, Richard C., Thomas H. Burbine, and Ian A. Franchi. "Linking asteroids and meteorites to the primordial planetesimal population." *Geochimica et Cosmochimica Acta* 277 (2020): 377-406.

Maurel, Clara, et al. "A Long-Lived Planetesimal Dynamo Powered by Core Crystallization." *Geophysical Research Letters* 48.6 (2021): e2020GL091917

Oran, Rona, Benjamin P. Weiss, and Ofer Cohen. "Were chondrites magnetized by the early solar wind?." *Earth and Planetary Science Letters* 492 (2018): 222-231.

Vernazza, Pierre, et al. "The formation and evolution of ordinary chondrite parent bodies." *Asteroids IV*. University of Arizona Press, 2015. 617-634.

Reviewer #3 (Remarks to the Author):

The manuscript by Wakita et al. explores the origin of intense magnetic anomalies that are located at the antipodes of numerous major impact basins on the Moon. The paper utilizes 3-D impact modeling using the iSALE hydrocode package to demonstrate that oblique basin-forming impacts are capable of depositing a significant amount metal-rich impactor-derived ejecta at basin antipodes. Some of this ejecta would be emplaced above the Curie temperature and could record sufficient thermoremanent magnetization upon cooling in a steady dynamo field to explain the intensity of the observed crustal remanence as inferred from spacecraft magnetic field measurements. The study was well done overall, and the revised version of the manuscript sufficiently addresses my previous comments as well as those of other previous reviewers and is now suitable for publication. This work is a welcome addition to the lunar magnetism literature as it provides a compelling explanation for at least one class of enigmatic lunar crustal magnetic anomalies.

Responses to the Reviewer #2's Comments:

The authors have addressed all outstanding issues from the previous review satisfactorily, and added simulations, plots, and text to support their findings. As such, I highly recommend the publication of this work.

There is one location where new added text needs to be more carefully stated. The line in question is (line 274), which states:

"While the impactor formed within 1-2 Myr after the birth of the Solar System could be fully differentiated like iron meteorites, the later formed bodies could be partially differentiated or remain undifferentiated like ordinary chondrites"

This needs to be stated more carefully. First, a distinction has to be made between meteorites and their parent bodies. Iron meteorites, or chondritic meteorites are not by themselves differentiated or not, but rather it is their parent bodies that may be so. Second, both may samples of different layer within a partially differentiated bodies (Elkins-Tanton et al. 2011,) and as of yet, there is no conclusive answer to whether or not chondrites come from undifferentiated parent bodies or whether iron meteorites come from fully differentiation parent bodies. In fact, several lines of evidence suggest partial differentiated asteroids may be the parent bodies of chondrites (Carpözen et al. 2011, Oran et al. 2018, Bryson et al. 2019) and well as some iron meteorites (Maurel et al. 2020). In general, the identification of meteorite parent bodies and their histories are still vigorously investigated (Vernazza et al. 2015, Greenwood et al. 2020)

I suggest editing this sentence, to both more accurately present the current state of knowledge of meteorite parent bodies, and also to clarify the context of this new text - and in particular how does the age and state of differentiation of possible impactors fits into the general hypothesis being tested here.

Reply: We thank the reviewer for this suggestion and rewrote this text as below.

“Large impactors like those considered here may be intact parent bodies or represent a piece of an even larger parent body. The formation time of the parent body helps determine whether it is differentiated or not [Hevey and Sanders 2006; Gail et al. 2014], which may result in a diversity of X_{TRM} values (see main text). When a parent body forms within 1-2 Myr after the birth of the Solar System, it may fully differentiate, like the parent bodies of iron meteorites [Krujier et al. 2013]. Later formed bodies could remain undifferentiated like a parent body of chondrites [Gail et al. 2014; Blackburn et al. 2019]. The parent body could also be partially differentiated, possibly producing both iron meteorites [Maurel et al. 2021] and chondrites [Carpözen et al. 2011; Elkins-Tanton et al. 2011; Bryson et al. 2019]. ”

Responses to the Reviewer #3's Comments:

The manuscript by Wakita et al. explores the origin of intense magnetic anomalies that are located at the antipodes of numerous major impact basins on the Moon. The paper utilizes 3-D impact modeling using the iSALE hydrocode package to demonstrate that oblique basin-forming impacts are capable of depositing a significant amount metal-rich impactor-derived ejecta at basin antipodes. Some of this ejecta would be emplaced above the Curie temperature and could record sufficient thermoremanent magnetization upon cooling in a steady dynamo field to explain the intensity of the observed crustal remanence as inferred from spacecraft magnetic field measurements. The study was well done overall, and the revised version of the manuscript sufficiently addresses my previous comments as well as those of other previous reviewers and is now suitable for publication. This work is a welcome addition to the lunar magnetism literature as it provides a compelling explanation for at least one class of enigmatic lunar crustal magnetic anomalies.

Reply: We are very grateful for your evaluation of our paper.

REVIEWERS' COMMENTS

Reviewer #2 (Remarks to the Author):

The authors have addressed all comments; the manuscript is of high quality and relevance and I recommend it for publication.